# Multi-dimensional optical information acquisition based on a misaligned unipolar barrier photodetector

Shukui Zhang[1,2,6], Hanxue Jiao[1,6], Yan Chen[1,3] ✉, Ruotong Yin[1], Xinning Huang[1], Qianru Zhao[1], Chong Tan[1], Shenyang Huang [4], Hugen Yan [4], Tie Lin[1], Hong Shen [1], Jun Ge[1], Xiangjian Meng [1], Weida Hu [1,2], Ning Dai [1,2], Xudong Wang [1] ✉, Junhao Chu[1,3] & Jianlu Wang [1,2,3,5] ✉

Acquiring multi-dimensional optical information, such as intensity, spectrum, polarization, and phase, can significantly enhance the performance of photodetectors. Incorporating these dimensions allows for improved image contrast, enhanced recognition capabilities, reduced interference, and better adaptation to complex environments. However, the challenge lies in obtaining these dimensions on a single photodetector. Here we propose a misaligned unipolar barrier photodetector based on van der Waals heterojunction to address this issue. This structure enables spectral detection by switching between two absorbing layers with different cut-off wavelengths for dual-band detection. For polarization detection, anisotropic semiconductors like black phosphorus and black arsenic phosphorus inherently possess polarization-detection capabilities without additional complex elements. By manipulating the crystal direction of these materials during heterojunction fabrication, the device becomes sensitive to incident light at different polarization angles. This research showcases the potential of the misaligned unipolar barrier photodetector in capturing multi-dimensional optical information, paving the way for next-generation photodetectors.

Infrared (IR) photodetectors play a crucial role in various fields, including night vision, astronomy, optical communications, health monitoring, and remote sensing[1]. The development of IR photodetectors is currently focused on integration and intelligence. To achieve this objective, it is essential to capture multi-dimensional optical information (such as intensity, polarization, wavelength, etc) of the target using a single photodetector. Obtaining more comprehensive target information provides data support for intelligent information processing and enables the rapid identification of targets,

including stealth and weak signals. However, integrating spectral detection and polarization detection into a single photodetector for traditional IR photodetectors presents challenges in terms of process and performance. Most reported photodetectors can only acquire optical information in a single dimension, be it intensity[2–5], polarization[6–10], or wavelength[11–15].

In recent years, there have been attempts to obtain multi-dimensional optical information using photodetectors based on artificial metasurface structures and two-dimensional (2D) materials[16–21].

[1]State Key Laboratory of Infrared Physics, Shanghai Institute of Technical Physics, Chinese Academy of Sciences, Shanghai, China. [2]Hangzhou Institute for Advanced Study, University of Chinese Academy of Sciences, Hangzhou, Zhejiang, China. [3]Institute of Optoelectronics, Shanghai Frontier Base of Intelligent Optoelectronics and Perception, Fudan University, Shanghai, China. [4]State Key Laboratory of Surface Physics and Department of Physics, Fudan University, Shanghai, China. [5]Frontier Institute of Chip and System, Fudan University, Shanghai, China. [6]These authors contributed equally: Shukui Zhang, Hanxue Jiao. ✉e-mail: yanchen_@fudan.edu.cn; wxd0130@mail.sitp.ac.cn; jlwang@mail.sitp.ac.cn

For instance, a dispersive Jones matrix method has been employed to construct a wavelength-decoupled coherent photodetector based on all-silicon metasurfaces, enabling the engineering of eigen-polarizations at different wavelengths[18]. However, the fabrication of such devices is complex, and they can only capture the polarization state of light at predefined discrete wavelengths. Additionally, a twisted double bilayer graphene photodetector has been developed to achieve simultaneous full-Stokes polarimetry and wavelength detection through tunable moiré quantum geometry. Nevertheless, this implementation requires a complex convolutional neural network trained on extensive datasets[17]. The current trend in multi-band photodetectors is to integrate spectral selectivity into a single pixel without the need for external mechanical and optical components. Similarly, the development trend in polarization detection aims to reduce the complexity of device surface structures and external mechanical components. Integrating multi-band detection and polarization detection into a single photodetector poses a highly challenging task.

Current dual-band photodetectors, such as mercury cadmium telluride (HgTeCd or MCT)[22,23], quantum well infrared photodetectors (QWIPs)[24,25], and antimony-based type-II superlattices (T2SLs)[26,27], are fabricated using multi-layered materials structure. The lattice mismatch issue during heteroepitaxial growth of these materials affects device performance. In recent years, there have been numerous reports on dual-band detectors prepared using novel materials without lattice mismatch problem, such as the sequential mode (bias-selectable) dual-band detector fabricated with mercury telluride (HgTe) quantum dots[28] and the simultaneous mode dual-band detector fabricated with silicon (Si)/molybdenum disulfide ($MoS_2$)/black phosphorus (bP) hybrid heterostructures[29]. However, the mature materials and quantum-dot dual-band IR photodetectors lack the capability of direct polarization detection. Integrating spectral and polarization detection in a single photodetector is highly challenging. 2D materials have the potential to achieve this purpose. Layered 2D materials, with their unique physical properties, hold great promise for optoelectronic applications[30–32]. These materials are composed of weak out-of-plane van der Waals bonds and lack dangling bonds on their surfaces after exfoliation. Consequently, different 2D materials can be combined to form van der Waals heterojunctions (vdWH), regardless of lattice mismatch, resulting in high-quality mutant heterojunctions[33,34]. Furthermore, the band gap of various 2D materials spans from ultraviolet to terahertz wavelengths[35,36], and 2D materials can also control the band gap by controlling the thickness of the material at the same time[2,34], enabling the design of devices with diverse band structures. Notably, photodetectors based on anisotropic 2D materials can achieve polarization detection without additional surface structures or external polarization optical elements[10,37–39].

There have been reports on biased switchable spectral/polarization photodetectors based on bP or black arsenic phosphorus (b-AsP). These photodetectors utilize bP/$MoS_2$/bP vdWH[10] and b-AsP/tungsten sulfide ($WS_2$)/b-AsP vdWH[21] to switch bias polarity, thereby achieving different polarization angle resolution detection. The pnp vdWH based on bP can achieve different spectrum detection by switching the bias voltage[14]. Furthermore, acquiring spatial polarization information can be achieved by utilizing twisted bP stacking[20]. Therefore, by utilizing the structure of a bias-switchable selective response channel and anisotropic 2D materials with different absorption cutoff wavelengths, we can develop a photodetector that obtains spectrum and polarization multi-dimensional light information on a single pixel.

In this work, we propose a misaligned unipolar barrier photodetector (MUBP) for acquiring multi-dimensional optical information. For spectral detection, we employ a two-terminal barrier structure with precise band engineering to enable bias-switchable dual-band detection. Compared to three-terminal devices, the two-terminal devices eliminate the need for an additional common electrode, resulting in simpler device structure and fabrication processes, higher fill factor, and improved device resolution[40]. It is important to note that the device, based on barrier structures, represents a new type of photodetector beyond the traditional pn junction, which has been extensively investigated in recent years for high-performance[41–45] or multi-band[23,26,27,46,47] IR detection. The unipolar barrier effectively blocks majority carrier transport while allowing minority carrier transport, leading to suppressed dark current without impeding photocurrent[48]. For polarization detection, we utilize anisotropic semiconductor bP and b-AsP as the absorbing layers, separated by a $MoS_2$ barrier layer. Furthermore, during the process of stacking heterogeneous junctions, we twist the crystal orientations of bP and b-AsP at a specific angle, rendering the two absorbing layers sensitive to incident light with different polarization angles. As a result, our MUBP enables the detection of multi-dimensional optical information without the need for additional metasurface nanostructure or complex mechanical components.

## Results

### Device design and working principle

Figure 1a illustrates the band structure of b-AsP, $MoS_2$, and bP before contact. The conduction band and valence band information of these materials has been obtained from previous reports[45,49]. The vdWH we designed, consisting of b-AsP/$MoS_2$/bP, follows a typical pBp barrier structure (p-type absorber, barrier layer, and p-type contact layer). This structure exhibits a large band offset at the valence band and an almost zero band offset at the conduction band, resulting in the blocking of hole transport while allowing electron transport[48]. Figures 1b, c illustrate the principle of bias-switched dual-band detection in the device. Under forward bias, the photogenerated carriers from bP with a cutoff wavelength of $\lambda_1$ can be collected to form a photocurrent, while those from b-AsP cannot be collected due to the barrier (Fig. 1b). The holes generated in b-AsP are impeded by the barrier, and although the electrons produced in b-AsP can be collected by the external circuit, the obstructed holes in b-AsP ultimately recombine. This indicates that the obstructed holes in b-AsP will recombine with an equal number of electrons, thus the electron-hole pairs generated in b-AsP do not contribute to the current. Under reverse bias, the opposite occurs, and the photocarriers from b-AsP with a cutoff wavelength of $\lambda_2$ are collected (Fig. 1c). Thus, the barrier photodetector is capable of dual-band detection depending on the polarity of the bias voltage. Figure 1d depicts the architecture of the b-AsP/$MoS_2$/bP barrier photodetector. Figure 1e, f demonstrate how the MUBP is sensitive to incident light at different polarization angles under different polarity bias voltages. During the stacking of heterojunctions, the crystal orientation of bP and b-AsP are twisted, enabling bP and b-AsP to be sensitive to different linearly polarized light. Consequently, the device is sensitive to the polarization of incident light at polarization angle $P_1$ (along the bP crystal direction) under forward bias (Fig. 1e) and at polarization angle $P_2$ (along the b-AsP crystal direction) under reverse bias (Fig. 1f).

The properties of b-AsP, $MoS_2$, and bP field effect transistor (FET) have been characterized (Supplementary Fig. 1). The pBp vdWH is fabricated by dry transfer stacking after mechanical exfoliation to obtain the required materials. Following the deposition of electrodes, a thin layer of h-BN is transferred to the top as a passivation layer. Cross-section high-resolution transmission electron microscopy (TEM) and energy-dispersive electron spectroscopy (EDS) images are captured at the b-AsP/$MoS_2$/bP vdWH interface (Supplementary Fig. 2). Further details of the fabrication process can be found in the Methods section. For this experiment, two types of b-AsP with different percentages of As atoms, namely b-As$_{0.6}$P$_{0.4}$ and b-As$_{0.83}$P$_{0.17}$, are used. The crystal structures of bP and b-AsP with varying arsenic atomic ratios are verified using Raman spectroscopy (Supplementary

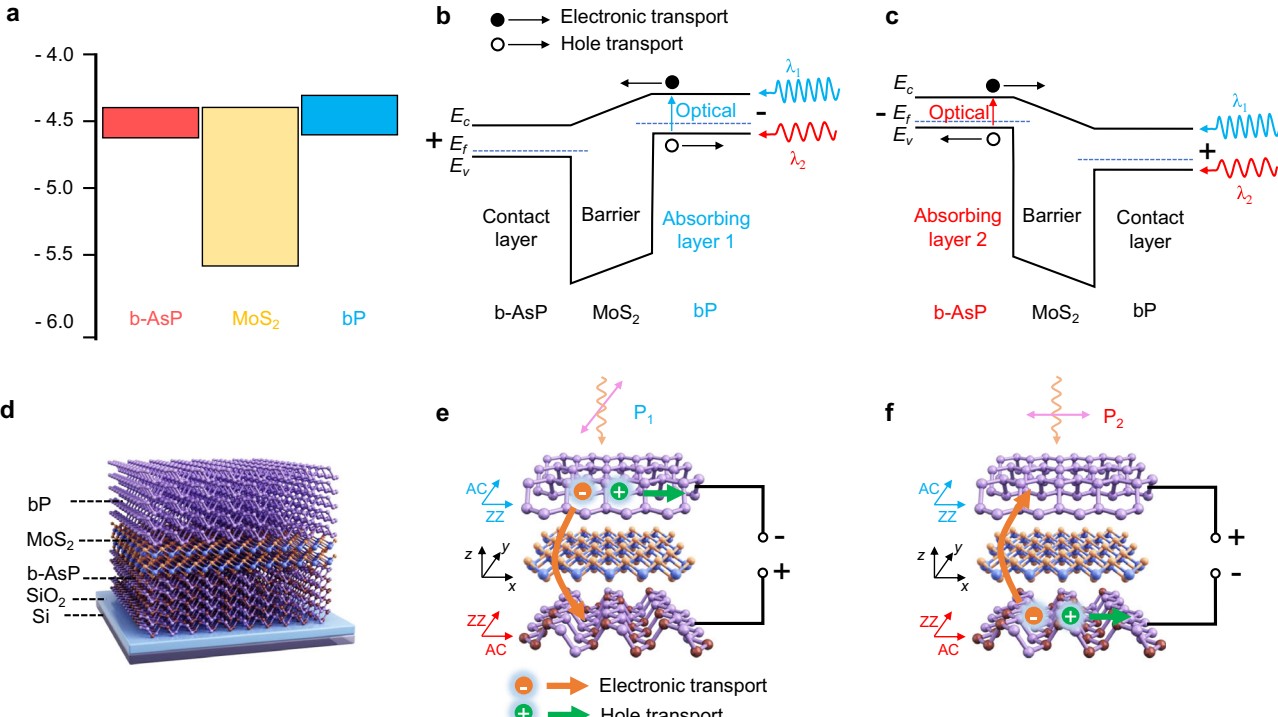

**Fig. 1 | Design of the misaligned unipolar barrier photodetector. a** Energy band profiles of black arsenic phosphorus (b-AsP), molybdenum disulfide (MoS$_2$), black phosphorus (bP) before equilibrium. **b, c** Energy band diagrams for misaligned unipolar barrier photodetector under forward and reverse bias, respectively. Under forward bias (reverse bias), the photogenerated carriers from absorbing layer bP (b-AsP) with a cutoff wavelength of $\lambda_1$ ($\lambda_2$) can be collected to form a photocurrent. $E_c$, $E_v$, and $E_f$ are the conduction band, valence band, and Fermi level, respectively.

**d** Schematic of the device, composed of vertically stacked b-AsP/MoS$_2$/bP heterostructure. **e, f** Schematic diagrams for misaligned unipolar barrier photodetector under forward and reverse bias, respectively. AC and ZZ axes correspond to the armchair and zigzag direction of crystal structure. The device is sensitive to the polarization of incident light at polarization angle P$_1$ (along the AC direction of bP) under forward bias, and at polarization angle P$_2$ (along the AC direction of b-AsP) at the reverse bias.

Fig. 3a). Supplementary Fig. 3b presents the polarization transmission spectra of bP and b-AsP, measured using Fourier transform infrared spectroscopy (FTIR). From Supplementary Fig. 3b, it can be observed that both bP and b-AsP exhibit anisotropic optical absorption characteristics, with the absorption cutoff edge of b-AsP shifting towards longer wavelengths as the percentage of arsenic atoms increases, which is consistent with previous reports[38,50]. bP has a band gap of 0.3 eV, while the band gap of b-AsP can be adjusted from 0.3 eV to 0.15 eV by adjusting the proportion of arsenic atoms[51]. This characteristic positions bP and b-AsP as promising candidates for mid-wavelength infrared (3-5 μm, MWIR) photodetectors.

To investigate the voltage-addressing behavior of MUBP, we conducted photocurrent mapping under different bias voltages. Figure 2 illustrates the operational mechanism of barrier structure dual-band detection, which utilizes forward/reverse bias voltages to select different response channels. In Fig. 2a, an optical image of the device is shown, with the contours of b-AsP, MoS$_2$, and bP highlighted by red, orange, and blue dashed lines, respectively. The schematic diagram of the device structure is displayed in the upper right portion of Fig. 2a, with b-AsP, MoS$_2$, and bP stacked from bottom to up. This device can be divided into three regions based on the type of junction area: Region I represents the b-AsP/MoS$_2$ junction area, region II represents the MoS$_2$/bP junction area, and region III represents the pBp vdWH composed of b-AsP/MoS$_2$/bP, as depicted in Fig. 2b–d, respectively.

To understand the underlying mechanisms of barrier structure photoresponse at different bias voltages, we obtained photocurrent mapping images at bias voltages of 0.5 V and −0.5 V, as shown in Fig. 2e, i, respectively. These measurements were performed under the illumination of an 830 nm laser, with electrode A1 as the source and electrode B2 as the drain. It is evident that under a forward bias voltage

of 0.5 V, the photoresponse originates from bP (Fig. 2e). The band structure schematics of regions I, II, and III under a bias of 0.5 V are presented in Fig. 2f–h, respectively. The internal electric fields of the b-AsP/MoS$_2$ junction 1 and MoS$_2$/bP junction 2 are denoted as $E_{in1}$ and $E_{in2}$, respectively. The applied electric field generated by the bias voltage is denoted as $E_{ex}$.

When operating under a forward bias, junction 1 is forward biased and junction 2 is reverse biased. As the forward bias voltage increases to $|E_{ex}| > |E_{in1}|$, the photogenerated holes produced by b-AsP layer cannot be transported to the drain due to the presence of the barrier layer, leading in recombination with electrons and no photocurrent generation, as shown in Fig. 2f. On the other hand, the bP layer absorbs photons and produces photogenerated electron holes, which are separated by the electric field and eventually collected by the electrode to form a photocurrent, as shown in Fig. 2g. The band structure of pBp vdWH under the forward bias opening operating voltage is shown in Fig. 2h. In another word when the forward bias voltage increases to $|E_{ex}| > |E_{in1}|$, the holes generated in the b-AsP layer, whether photogenerated or thermally excited, will be blocked by the barrier layer, and the bias applied at this point can be defined as the operating voltage $V_{op}$ of junction 2 (bP channel, $\lambda_1$ cut-off wavelength). Therefore, the pBp barrier structure can not only reduce the electrical crosstalk of the dual-band detection but also suppress the dark current generated by the non-absorbing layer when operating at the opening voltage[46]. When the reverse bias reaches the open operating voltage, the situation reveres, as shown in Fig. 2i (b-AsP channel, $\lambda_2$ cut-off wavelength). The change of pBp heterojunction band structure is opposite to that under reverse bias, as shown in Fig. 2j–l, respectively. Thus, the bias-switchable two-channel response is obtained under two different bias polarities in the MUBP.

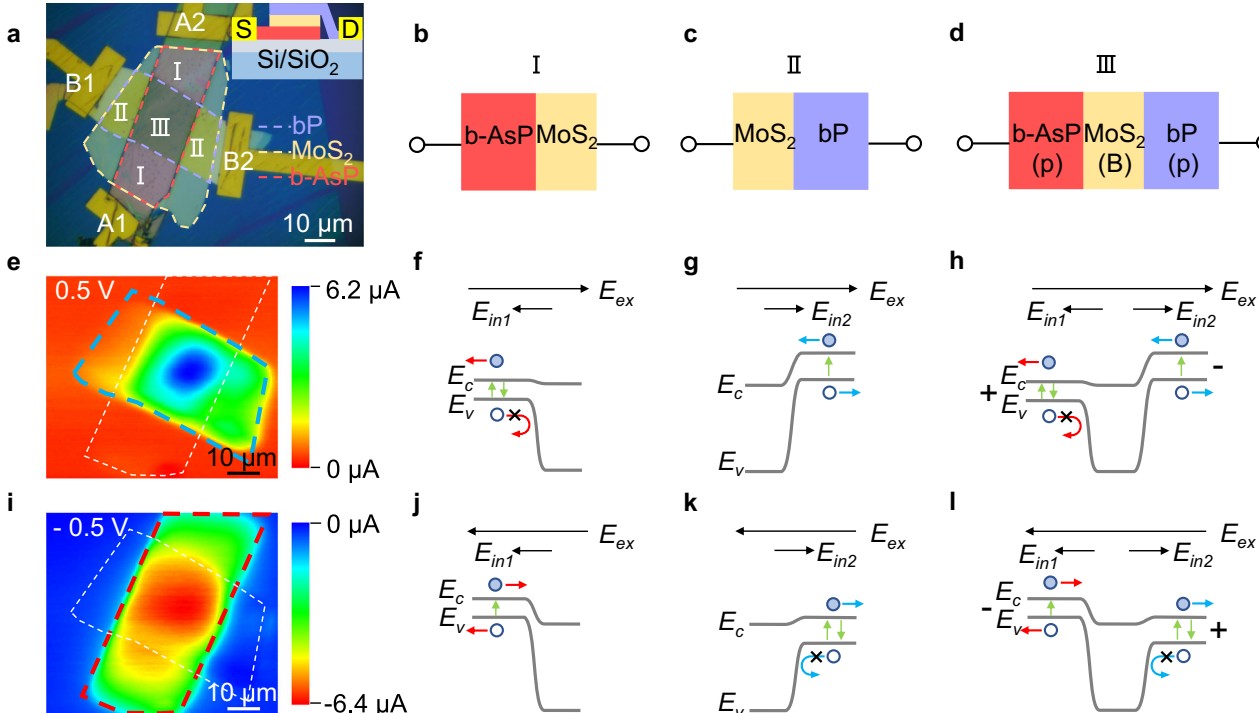

**Fig. 2 | The voltage-addressing behavior of b-AsP/MoS₂/bP unipolar barrier (pBp) photodetector. a** Optical microscope image of the device. The contours of b-AsP, MoS₂, and bP highlighted by red, orange, and blue dashed lines, respectively. A1, A2, B1, B2 are electrodes. Upper right corner: the diagram of the device structure, with b-AsP (red), MoS₂ (orange), and bP (blue) stacked from bottom to up. **b–d** Diagram of heterojunction corresponding to regions I, II, and III in **a**. **e** Photocurrent mapping at a forward bias voltage of 0.5 V. The contours of b-AsP and bP highlighted by white and blue dashed lines, respectively. **f–h** Energy band diagrams of corresponding regions I, II, and III at a forward bias voltage of 0.5 V. $E_{in1}$ and $E_{in2}$ are the internal electric fields of the b-AsP/MoS₂ junction 1 and MoS₂/bP junction 2, respectively. $E_{ex}$ is the applied electric field generated by the bias voltage. The black arrows represent the direction of the electric field, the blue (red) arrows represent the transport direction of the carriers in bP (b-AsP). **i** Photocurrent mapping at a reverse bias voltage of −0.5 V. The contours of b-AsP and bP highlighted by red and white dashed lines, respectively. **j–l** Energy band diagrams of corresponding regions I, II, and III at a reverse bias voltage of −0.5 V.

To investigate the transition states occurring before reaching the operating voltage during the switching between forward and reverse bias voltages, we analyzed the photocurrent mapping of the device at 0.1 V, 0 V, and −0.1 V bias (Supplementary Fig. 4). To provide a comprehensive understanding of the physical processes involved in these transition states and the analysis of reaching the open operating voltage $V_{op}$, we have included a detailed discussion in Supplementary Note 1. Additionally, Supplementary Fig. 5 displays the voltage-addressing photocurrent mapping of the device under incident light illumination of 1550 nm. Furthermore, Supplementary Fig. 6 and Supplementary Fig. 7 present the photocurrent mapping images of device 2 and device 3, respectively, under different bias voltages, providing additional evidence for the stable reliability of our photodetectors.

**Photodetector characterization**

To verify that the MUBP can achieve spectral detection and polarization detection, we use an FTIR spectrometer to characterize the polarization-resolved spectral response of the device under forward bias and reverse bias respectively at liquid nitrogen temperature. Device 4 is used here to characterize the FTIR response performance. The thicknesses of bP, MoS₂ and b-AsP in device 4 are 300 nm, 20 nm and 32 nm, respectively (Supplementary Fig. 8). The FTIR measurement configuration is shown in Supplementary Fig. 9a. For the polarization measurement, a ZnSe polarizer was introduced in front of the sample. In Fig. 3a, a comparison of the polarization spectral responses between the forward and reverse bias of the MUBP (b-As₀.₈₃P₀.₁₇/MoS₂/bP) is presented (The spectral response of a non-polarized light source

is shown in Supplementary Fig. 10). It can be seen that the cut-off wavelength $\lambda_1$ at a forward bias (channel 1, bP) is 4.2 μm, and the cut-off wavelength $\lambda_2$ at a reverse bias (channel 2, b-AsP) is 4.9 μm, and both channels exhibit polarization sensitive responses. Since the carbon dioxide (CO₂) has an absorption peak at 4.2 μm (see Supplementary Fig. 9b), there is a gap in the spectral response of reverse bias (Channel 2, b-AsP) at 4.2 μm. Depending on the location of the absorption peak of CO₂, MWIR can be divided into MWIR1 (shorter than 4.2 μm) and MWIR2 (longer than 4.2 μm). It can be seen that the spectral response range of channel 1 (bP) is MWIR1, while the spectral response range of channel 2 (b-AsP) covers MWIR1 + MWIR2 and the response of the MWIR1 is very similar to that of channel 1. Usually, the optical overlap of traditional dual-band photodetectors is not conducive to spectral resolution, but here we can distinguish the spectral information of MWIR1 and MWIR2 by using the same spectral response trend of bP and b-AsP in the MWIR1 range. When the normalized device detects the incident light of unknown wavelength, if the value of the device channel 2 signal minus channel 1 signal is zero, the wavelength of the incident light is in the MWIR1 range. If the value is greater than zero, the incident light is in the MWIR2 range (Supplementary Fig. 9c). For convenience, we define channel 1 as the MWIR1 channel and channel 2 as the MWIR2 channel in the following text. As a result, our photodetector is capable of discerning between MWIR1 and MWIR2 response spectra, enabling polarization detection within their respective spectral response ranges. Therefore, this MUBP, exhibiting distinct absorption spectra and anisotropic spectral response, offers a promising avenue for the advancement of multi-dimensional optical information photodetectors in the future.

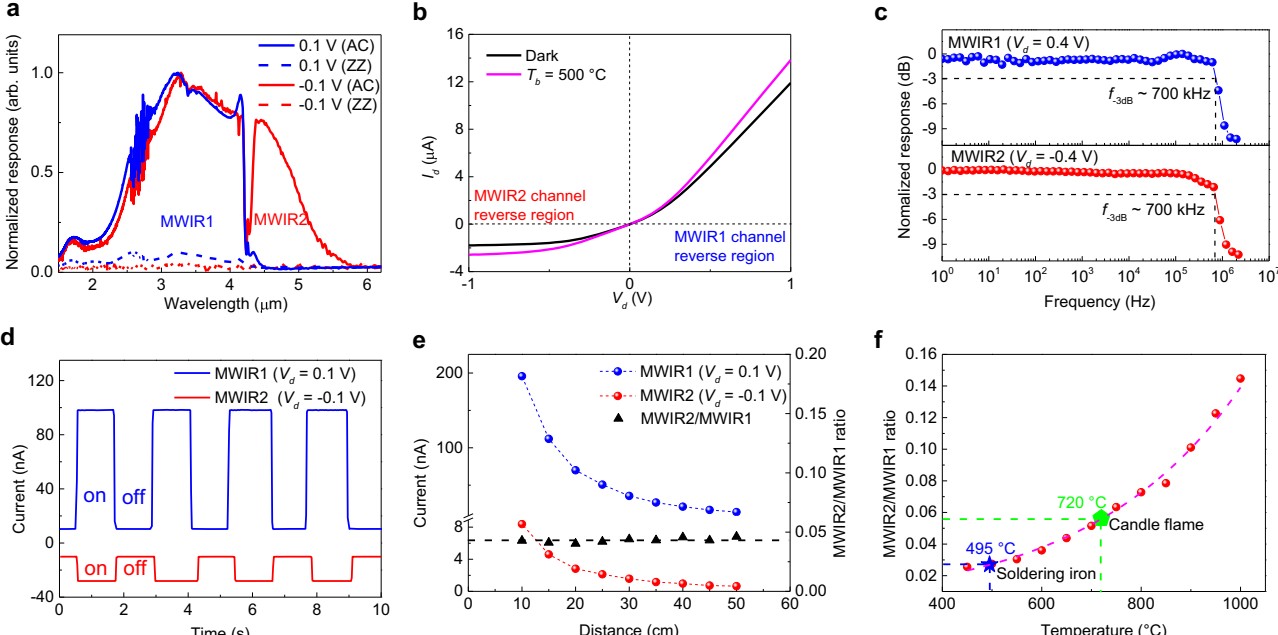

**Fig. 3 | Bias-switch dual-band detection. a** Relative response spectrum with polarization resolution of the mid-wavelength infrared (MWIR) photodetector. The blue solid line and dashed line are polarized incident light incident along the AC direction and ZZ direction of bP, respectively. The red solid line and dashed line are polarized incident light incident along the AC direction and ZZ direction of b-AsP, respectively. **b** I−V curves of the photodetector in the dark and under 500 °C blackbody radiation illumination. **c** Frequency response of MWIR1 channel and MWIR2 channel to 1550 nm modulated laser. $V_d$ is the bias voltage and $f_{-3dB}$ is the frequency at −3dB point. **d** Photoresponse of MWIR1 channel and MWIR2 channel to the blackbody radiation respectively. **e** Left axis, MWIR1 and MWIR2 signal as a function of detection distance. Right axis, the relationship between the MWIR2/MWIR1 ratio and detection distance. The horizontal black dashed line is the fitting line. **f** Summary of the MWIR2/MWIR1 ratio depends on the temperature of the detected target. Red dots show the measured MWIR2/MWIR1 ratio as a function of blackbody temperature and the pink dashed line is the fitting curve.

The detection performance of the photodetector for blackbody radiation at room temperature is evaluated. The measurement setup for blackbody radiation is illustrated in Supplementary Fig. 11a, with the photodetector placed in a vacuum Dewar. By utilizing 500 °C blackbody as the radiation source, the output characteristic curve of the MUBP is shown in Fig. 3b, demonstrating the sensitivity of our photodetector to blackbody radiation at room temperature. The effective incident blackbody radiation power can be calculated using the equation[52]: $P_{bb} = \eta \frac{\sigma(T_b^4 - T_0^4)A_b A_D}{2\sqrt{2}\pi L^2}$, where $\eta = 0.7$ is the transmittance of Dewar window, σ is the Stefan-Boltzmann constant, $T_b$ is the blackbody temperature, $T_O$ is the operating temperature of the photodetector, $A_b$ is the area of the blackbody radiation aperture and $L$ is the distance from the blackbody to the detector. $A_D$ is the area of the active region of the detector (as shown in Fig. 2e, f, the regions of forward and reverse bias responses for the device are II + III regions and I + III regions, respectively. Here, the active region of the detector under forward bias and reverse bias for our sample are 740 μm² and 1300 μm², respectively). When the blackbody temperature is 500 °C, the blackbody radiation power density at a distance of 10 cm from the blackbody radiation source can be calculated as $7.9 \times 10^{-11}$ W μm⁻². The blackbody responsivity $R_{bb}$ given by $R_{bb} = I_{ph}/P_{bb}$. Based on the data extracted from Fig. 3b, we can calculate the blackbody responsivity of the MWIR1 channel ($V_d = 0.1$ V) and MWIR2 channel ($V_d = -0.1$ V) to be 857 mA/W and 519 mA/W, respectively.

The noise spectrum of the device from 1 to $10^5$ Hz at room temperature is illustrated in Supplementary Fig. 12. It is observed that the noise behavior is characterized as 1/f noise at low-frequency. Since our device bandwidth is 700 kHz (as shown in Fig. 3c), we calculated the detectivity using the noise current of the device at 10 kHz to 100 kHz. The noise current, denoted as $I_n$, was estimated to be $1 \times 10^{-13}$ A Hz⁻¹ᐟ² ($V_d = 0.1$ V) and $3 \times 10^{-13}$ A Hz⁻¹ᐟ² ($V_d = -0.1$ V), respectively.

The specific detectivity $D^*$ is a crucial parameter for evaluating the performance of IR photodetectors and can be determined using the following equation[52]:

$$D^* = \frac{\sqrt{A_D \Delta f}}{NEP} = \frac{R\sqrt{A\Delta f}}{I_n} \quad (1)$$

where $A_D$ is the effective area of the photodetector, $\Delta f$ is the bandwidth, $R$ is the responsivity. By applying this equation, the specific detectivity $D^*$ of the MWIR1 and the MWIR2 photodetector can be calculated as $2.3 \times 10^{10}$ cmHz¹ᐟ²W⁻¹ ($V_d = 0.1$ V) and $6.2 \times 10^9$ cmHz¹ᐟ²W⁻¹ ($V_d = -0.1$ V), respectively.

Moreover, we conducted an investigation into the speed of the dual-band photodetector to a modulated laser at room temperature in the atmosphere. Supplementary Fig. 13 showcases the response of the MWIR1 photodetector and MWIR2 photodetector to the modulated laser with a wavelength of 1550 nm. Notably, both photodetectors exhibit a remarkably fast response to the laser, with rise and fall times of approximately 0.5 μs. Furthermore, Fig. 3c displays the frequency response of the MWIR1 photodetector and MWIR2 photodetector to a 1550 nm modulated laser, with both photodetectors reaching their −3 dB points at ~700 kHz.

## Temperature measurement with colorimetric method
Dual-band IR photodetectors are more accurate and reliable than monochrome IR photodetectors in remote temperature measurement[28,29,40]. According to Wien's displacement law, the product of the absolute blackbody temperature and the wavelength corresponding to the maximum radiation intensity is a constant ($\lambda T = b$). This implies that as the temperature of an absolute blackbody increases, the maximum radiation power shifts towards the shorter wavelengths (as shown in Supplementary Fig. 11b). In Planck's law

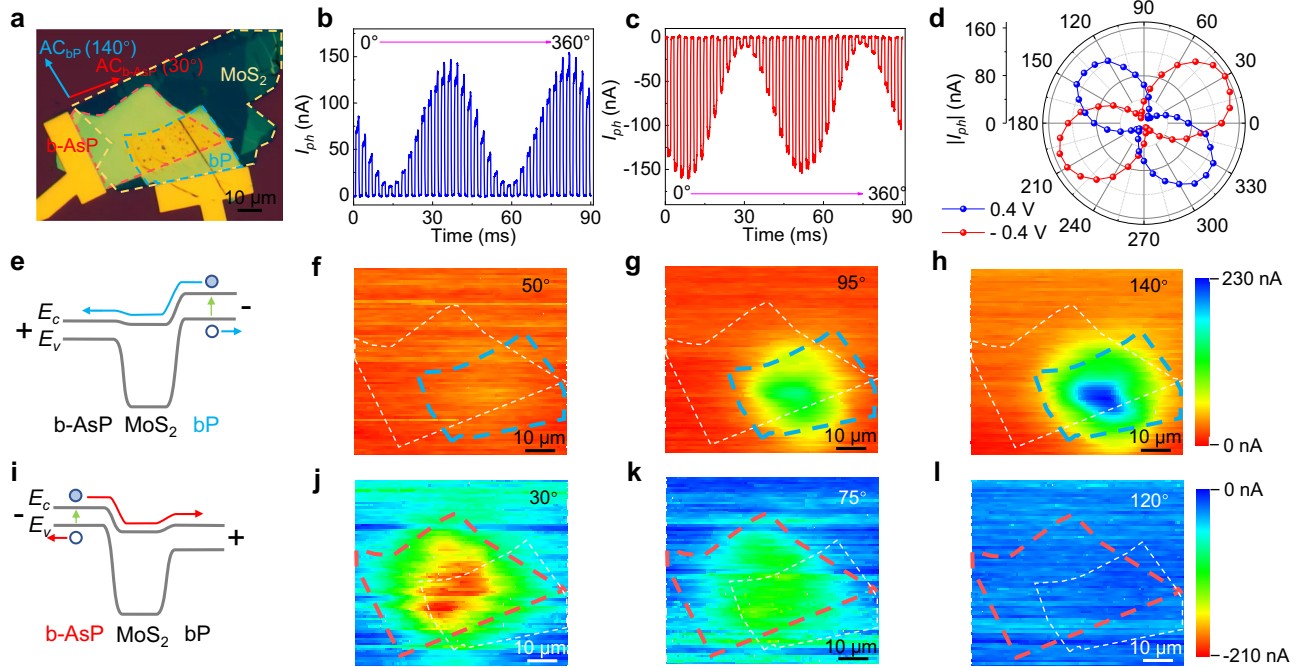

**Fig. 4 | Switchable polarization resolution detection. a** Optical microscope image of the device. The contours of b-AsP, MoS$_2$, and bP highlighted by red, orange, and blue dashed lines, respectively. The AC$_{bP}$ and AC$_{b-AsP}$ are the AC directions of bP and b-AsP, respectively. **b, c** The photocurrent of the device at 0.4 V and −0.4 V with the polarization angle of the linearly polarized incident light sweeping from 0° to 360°. The scanning step of polarization angle change after each photocurrent switch change is 10°. **d** Polar plot of forward and reverse bias photocurrents varying from 0° to 360° in the polarization angle of incoming ray

polarized light. The polarization angle of 0° is aligned along with the horizontal direction in **a**. **e** Energy band diagram of the device under forward bias. **f–h** Photocurrent mapping images of linearly polarized light with incident angles of 50°, 95°, and 140° at a forward bias of 0.4 V. **i**, Energy band diagram of the device under reverse bias. The contours of b-AsP and bP highlighted by white and blue dashed lines, respectively. **j–l** Photocurrent mapping images of linearly polarized light with incident angles of 30°, 75°, and 120° at a negative bias of −0.4 V. The contours of b-AsP and bP highlighted by red and white dashed lines, respectively.

under Wien's approximation, the emission power of the target can be expressed as[29]:

$$M = \frac{2\pi h \varepsilon b \lambda^{-5} \Delta\lambda}{e^{\frac{hc}{k_B \lambda T}}} \qquad (2)$$

Here, $h$ is Planck's constant, $\varepsilon$ is the emissivity, $b$ is the Wien constant, $\lambda$ is the wavelength, $\Delta\lambda$ is the wavelength width, $c$ is the speed of light, and $T$ is the temperature of the object. When the target being measured behaves as a blackbody, the temperature obtained from the target is both accurate and reliable. However, if the target behaves differently from a blackbody, knowledge of the target object's emissivity becomes essential for accurate data acquisition from a monochrome photodetector. In contrast, dual-band photodetectors eliminate the necessity of knowing the object's emissivity, resulting in more precise and reliable temperature measurements compared to monochrome photodetectors.

It determines the temperature of an object based on the ratio of the radiation energy in two adjacent bands of the object's IR radiation[29]. The ratio can be expressed as:

$$Q = \frac{M_1}{M_2} = \frac{\varepsilon_1 \Delta\lambda_1}{\varepsilon_2 \Delta\lambda_2} \left(\frac{\lambda_2}{\lambda_1}\right)^5 exp\left[\frac{hc}{k_B T}\left(\frac{1}{\lambda_2} - \frac{1}{\lambda_1}\right)\right] \qquad (3)$$

$$lnQ = C_1 + C_2 T^{-1} \qquad (4)$$

In this context, $C_1$ and $C_2$ represent the system constants. It is worth noting that colorimetric temperature measurement remains

unaffected by emissivity, rendering it suitable for absolute and remote temperature measurement applications.

As shown in Fig. 3d, the photoresponse of the MWIR1 channel and the MWIR2 channel to the blackbody source is depicted, with the Dewar operating at liquid nitrogen temperatures to ensure a favorable signal-to-noise ratio. Notably, Fig. 3e demonstrates that the signal ratio of MWIR2/MWIR1 in our dual-band photodetector remains relatively constant despite variations in detection distance under a constant blackbody temperature. This finding suggests that colorimetric temperature measurement remains unaffected by changes in distance. Furthermore, the relationship between the MWIR2/MWIR1 signal ratio and changes in blackbody temperature at a constant detection distance was examined (Fig. 3f). It was observed that as the blackbody temperature increased from 450 °C to 1000 °C, the MWIR2/MWIR1 signal ratio correspondingly increased from 0.026 to 0.145. By utilizing the blackbody response as a reference standard, remote temperature measurements of a soldering iron and a candle flame were conducted. The temperature of the electric soldering iron was measured at 495 °C, which closely aligned with the actual set temperature of 500 °C, within the system's error tolerance. Similarly, the temperature of the candle flame was measured at approximately 720 °C. These results serve as evidence that our dual-band photodetector can effectively be employed for remote temperature measurement of various target objects. Notably, even when the blackbody response frequency of the photodetector reaches 500 Hz, the MWIR2/MWIR1 ratio remains constant, as depicted in Supplementary Fig. 11c. Furthermore, considering that the blackbody radiation power is associated with the detection distance, the relationship between the responsivity and blackbody radiation power can be derived from Fig. 3e, as illustrated in Supplementary Fig. 14. This highlights the

capability of our photodetector to detect light intensity information in addition to its dual-band functionality.

### Bias-switchable polarization-resolved detection

To demonstrate the bias-switchable polarization-resolved detection, we investigated the response of a heterojunction composed of b-As$_{0.6}$P$_{0.4}$/MoS$_2$/bP to linear polarized light after switching different bias voltages. The optical photo of the device is presented in Fig. 4a. As shown in Fig. 4b, c, the relationship between the periodic increase and decrease of the photocurrent is depicted as the polarization angle of the incident light ($\lambda$ = 4.6 μm) changes from 0° to 360° at a bias voltage of 0.4 V and −0.4 V, respectively. The measured polarization extinction ratios were found to be 24.7 and 11.8 for the bias voltages of 0.4 V and −0.4 V, respectively. Furthermore, Fig. 4d illustrates the polar plot of forward and reverse photocurrents, determined by polarization angles of the incident light. It is evident that as the polarization angle varies from 0° to 180°, the photocurrent under a bias of 0.4 V is smallest at 50° and largest at 140°. Conversely, under a bias of −0.4 V, the photocurrent is maximum at 30° and minimum at 120°. This result indicates that the AC direction of bP is 140°, while the AC direction of b-AsP is 30°, as shown by the blue and red arrows in Fig. 4a. The photocurrent and polarization angle at both 0.4 V and −0.4 V adhere to the functional relation[21]: $I_{ph} = a\cos(2\theta + \varphi) + b$, with a phase difference of 110°. The phase difference is determined by the twisted angle between the crystal direction of bP and b-AsP during the stacking of the vdWH.

The origin of heterojunction polarization detection is further confirmed through polarized photocurrent mapping. Figure 4e, i present schematic diagrams of the band structure when the bias voltage is 0.4 V and −0.4 V, respectively. Additionally, Fig. 4f–h display polarization-resolved photocurrent mapping images of the MWIR1 channel at polarization angles of 50°, 95°, and 140°, respectively, with the incident light wavelength set at 4.6 μm. It can be seen that the photocurrent is generated in the bP region of the device, and the photocurrent reaches its maximum at 140°. Similarly, Fig. 4–l depict the polarization-resolved photocurrent mapping images of the MWIR2 channel at polarization angles of 30°, 75°, and 120°, respectively. The device generates photocurrent in the b-AsP region, and the photocurrent is maximum at 30°. It is evident that the trend of the current changing with the polarization angle aligns with that shown in Fig. 4d. This observation further confirms the polarization recognition capability of the device with a bias switch. However, it is important to note that the method described above for distinguishing polarization states is only suitable for light with a wavelength within the overlapping absorption region of the bP and b-AsP spectra. Specifically, it applies to linearly polarized incident light with a wavelength below 4.2 μm. For the identification of polarization states of light above 4.2 μm, at least two photodetectors are necessary to achieve accurate identification.

### Discussion

In summary, we have designed and fabricated a MUBP utilizing a b-AsP/MoS$_2$/bP vdWH for acquiring multi-dimensional optical information. The optical information of intensity, spectrum, and polarization can be captured by this single photodetector. It incorporates two distinct absorption layers, the MWIR1 channel (bP) and the MWIR2 channel (b-AsP), separated by a MoS$_2$ barrier layer. This arrangement effectively suppresses the dark current while enabling efficient photocurrent generation. The dual-band response is achieved by altering the polarity of the applied bias and realized temperature measurement. Furthermore, the device exhibits bias-selective polarization angle resolution detection characteristics, with the dichroism extinction ratio reaching 24.7 for the MWIR1 channel and 11.8 for the MWIR2 channel. Our research proposes a practical approach for integrating multi-band detection and polarization detection into a single photodetector. The multi-dimensional optical

information photodetector has great potential and more applications in the field of photon detection.

## Methods

### Device fabrication

The b-AsP, MoS$_2$, and bP used in this experiment are obtained by mechanical exfoliation method, and the b-AsP/MoS$_2$/bP vdWH was stacked on a silicon substrate with 285 nm SiO$_2$ by the fixed-point transfer method on a self-constructed transfer platform. To prevent material oxidation, these processes are carried out in a nitrogen-filled glove box. The electrode patterns were obtained by electron beam lithography, and then the metal films Cr/Au (20/80 nm) were successively deposited by thermal evaporation. After the lift-off process, the electrodes were obtained. Finally, a thin h-BN was transferred to cover the device to isolate the air and water. All 2D materials were purchased from HQ Graphene.

### Photodetector characterization

The photocurrent mapping images and response time were taken at room temperature and under ambient conditions by the MStarter 200 optoelectronic measurement system from Maita Optoelectronic Technology Co., LTD. The spectral response was measured with an FTIR spectrometer. The vacuum Dewar loaded with the device was placed at the auxiliary exit port of the spectrometer, and exciting illumination from the Glowbar source (1000 K blackbody) was focused on the sample using a ZnSe lens. To obtain a good spectral response, the device operates at liquid nitrogen temperatures. The bias voltage of the device was provided by a current amplifier (Stanford Research Systems SRS570), and the signal generated by the device was collected and analyzed by the computer. The background signal was obtained by the internal deuterated triglycine sulfate (DTGS) detector of the FTIR under identical measurement conditions. Polarization measurements were achieved by placing a ZnSe wire grid polarizer in the beam path. To obtain detectivity, blackbody radiation measurements are performed at room temperature. The vacuum Dewar was placed in front of the blackbody radiation source (Newport Oriel 67000) without a focusing lens, and a chopper was used to modulate the illumination. The photoresponse signal of the device was collected by the Agilent B2912 source meter. The noise current density was measured using a preamplifier (SR570) and dynamic signal analyzer (Keysight 35670 A) in the vacuum Dewar at room temperature.

## Data availability

The Source Data underlying the figures of this study are available with the paper. All raw data generated during the current study are available from the corresponding authors upon request. Source data are provided with this paper.

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

## Acknowledgements

This work is supported by the Strategic Priority Research Program of the Chinese Academy of Sciences (Grant No. XDB0580000 (X.W.)), Natural Science Foundation of China (Grant Nos. 62222413 (X.W.), 62025405 (J.W.), 62075228 (X.M.), 62105100 (Y.C.) and 62334001 (X.W.)), Key Research Program of Frontier Sciences, CAS (Grant No. ZDBS-LY-JSC045 (X.W.)), Strategic Priority Research Program of the Chinese

Academy of Sciences (Grant No. XDB44000000 (J.W.)), Natural Science Foundation of Shanghai (Grant No. 23ZR1473400 (X.W.)), Hundred Talents Program of the Chinese Academy of Sciences (X.W.), Hangzhou Science and Technology Bureau of Zhejiang Province (No. TD2020002 (N.D.)), and China Postdoctoral Science Foundation (2022M710154 (S.Z.)).

## Author contributions

J.W. conceived and supervised the research. S.Z., H.J., R.Y., X.H., Q.Z., T.C., and S.H. performed the device characterizations. T.L., H.S., X.M., and J.G. advised on the experiments and data analysis. W.H., H.Y., N.D., and J.C. provided experimental testing support. J.W. was responsible for project planning. S. Z. and H.J. analyzed the data and drafted the manuscript. X.W., Y.C., and J.W. revised the manuscript. All authors discussed the results.

## Competing interests

The authors declare no competing interests.
