## [Peer Review file · Nature Communications]

Multi-dimensional optical information acquisition based on a misaligned unipolar barrier photodetector

Corresponding Author: Professor Jianlu Wang

[Parts of this Peer Review File have been redacted as indicated to remove third-party material.]

[The author's responses to all Reviewer comments can be found at the end of this file]

Reviewer comments (first round of review):

Reviewer #1

(Remarks to the Author)

In this manuscript, the authors have developed an innovative twisted unipolar barrier photodetector composed of b-AsP/MoS₂/BP heterostructure for acquiring multi-dimensional optical information. For spectral detection, they employ a two-terminal barrier structure with precise band engineering to enable bias-switchable dual-band detection. For polarization detection, they utilize anisotropic semiconductor BP and b-AsP as the absorbing layers, separated by a MoS₂ barrier layer. Furthermore, during the process of stacking heterogeneous junctions, they twist the crystal orientations of BP and AsP at a specific angle, rendering the two absorbing layers sensitive to incident light with different polarization angles.

It is impressive enough that such a structure allows for dual-band detection and polarization detection simultaneously on only one pixel. I suggest to accept this manuscript. However, there are some issues that need to be addressed before the manuscript can be published.

1. In Figure 2, the authors used an 830 nm laser source for photocurrent mapping. It would be helpful to mention that previous studies have investigated the use of multilayer MoS₂ as photodetectors for light sources with a wavelength of approximately 900 nm. Can the MoS₂ layer also absorb light at 830 nm?
2. Please explain why carriers are not collected from outside the studied area.
3. In figure 3a, the photoresponse of the MWIR2 channel is very similar to that of the MWIR1 channel above the CO₂ line. Please explain why the light on the MWIR2 channel is not absorbed by the MWIR1 channel first?
4. When the device is illuminated and driven by reverse bias, the bottom b-AsP layer acts as an absorber, while the top bP layer acts as a polarizer for polarization detection. What effect does this have on the bottom b-AsP absorber layer?
5. Figure S11 shows that the responsivity depends on the optical power, but this is not discussed in the text. Please explain why the responsivity is not constant.

Reviewer #2

(Remarks to the Author)

In this work, the authors reported a twisted unipolar barrier photodetector based on bP/MoS₂/b-AsP heterojunctions, the device has a capability of bias-switchable dual-band detection due to the unipolar barrier structure. By employing the anisotropic semiconductors such as black phosphorus (bP) and black arsenic phosphorus (b-AsP) as two absorbing layers, the authors also achieved the polarization-detection capabilities with polarization ratio (PR) value of 24.7 and 11.8 for the bias voltages of 0.4 V and -0.4 V, respectively, in the mid-wave infrared range.

Although the as-fabricated twisted unipolar barrier photodetector is capable of acquiring multi-dimensional optical information (including intensity, spectrum and polarization), similar device architecture and results about this subject have been previously reported. For example, the unipolar-barrier van der Waals heterostructures based on twisted b-AsP/WS₂/b-AsP were reported [Adv. Mater. 2022, 34, 2203766], which also showcase a bias-switchable polarization-sensitive photo-detection with PR value up to 55 and 42 at bias of -0.1 and 0.1 V, respectively, at 4.6 μm, even outperforming the device

described in this work. In addition, the bias-selectable and dual-band infrared detectors were also recently reported [ACS Nano 2023, 17, 12, 11771; Nano Lett. 2022, 22, 8, 3425]. Therefore, this work may not bring new conceptual or technological advance over the previous related reports. Given this, the reviewer cannot recommend it for publication in Nature Commun. The following are some concerns that need to be addressed.

1. The authors fabricated a twisted unipolar barrier photodetector, what is the exact twist angle between bP and b-AsP? Whether the twist angle has influence on the performance?
2. How does the author get the energy band as shown in Figure 1a? The theoretical calculation or experiments should be performed, not just cited the previous reports.
3. On page 4: "while those from b-AsP cannot be collected due to the barrier (Figure 1b)." The barrier can block the transport of photo-generated holes, but the electrons from b-AsP should be collected by the external circuit. Does author explain this in more detail?
4. On page 4: "the thickness of MoS2 is confirmed to be less than 20 nm through atomic force microscopy (AFM). The thickness of the absorption layer can be adjusted as needed." How does author optimize the thickness of MoS2 and absorption layer? No related thickness-dependent experiments were found in the manuscript.
5. In the photocurrent mapping (Fig. 2e, i) under 830 nm laser, if the MoS2 components contribute to the formation of photocurrent? Why it is or not.
6. What is the noise current of the device? The noise spectral density should be measured.
7. The resolution and contrast of photocurrent mapping in Fig. 4f-i is poor, what is the reason.

Reviewer #3

(Remarks to the Author)

The authors have fabricated a b-P/MoS2/b-AsP MWIR photodetector and shown that they can detect both spectral and linear polarisation information from the incoming illumination source. The study builds on an existing body of literature from the authors and other groups; however, some aspects of the study are novel, and I believe it should eventually be published. There are several significant issues that need to be addressed beforehand, as detailed below.

Given the impressive room temperature detectivity values reported for unpolarised light (especially for the b-AsP on the bottom of the device underneath the thick b-P layer), the readership will want to see more information on the detectivity/responsivity calculations. For example, why do the blackbody responsivity values mentioned in the main text (~10 to 4 A/W) differ from the blackbody responsivity values provided in the SI Figure S11 (max 0.8 to 0.03 A/W)? at what power density were the champion detectivity results measured? What cut-off wavelengths are used to estimate the incident power from the blackbody for the b-P and b-AsP? What region of the device was used to estimate the active area? In addition, it would be better to measure the noise current directly.

The ability to detect both spectral and polarisation information in a single detector is presented as an advantage in the present study. However, this may not translate to a practical advantage. For example, based on Figure 3a, how would this detector discriminate between 3 μ m and 5 μ m linearly polarised light aligned to the AC direction of the bottom b-AsP absorber (given that the top b-P will not strongly respond to either of these)? According to the subtraction method in the main text, this would be interpreted as light above 4.2 μ m. Hence, while the detector can independently detect polarisation and spectral information, I don't think it can perform these simultaneously without a second detector/polariser. This should be clarified within the text.

In addition to the above comment, the spectra provided in Figure 3a represent the simplest extreme situation (i.e linear polarisation aligned to the two crystal axis). To support the claim that the detector can use a subtraction method (or similar) to provide spectral/polarisation information, the bias-switchable spectral behaviour should also be measured using an unpolarised source (or even as a function of linear polarisation angle). I suspect that such method would hold up well to this more realistic case. I suggest the authors perform such measurements if they want to make these claims.

The novelty of the study needs to be clarified and the existing literature acknowledged. For example, the concept of detecting linear polarisation using layers of anisotropic 2D materials stacked at right angle to one another in a bias-switchable form is presented as novel, however, this concept has been demonstrated previously. See for example Nature Photonics volume 12, pages 601–607 (2018) which uses a very similar b-P/MoS2/b-P structure to detect MWIR linear polarisation. This similar earlier study is not mentioned in the manuscript, which may mislead the readership about the specific novelty of the work. Similarly, more recent studies exploring bias-switchable spectral response of b-P/MoS2/other IR absorber devices have not been acknowledged, see for example: ACS Appl. Mater. Interfaces 2022, 14, 28, 32665–32674 and ACS Nano 2023, 17, 12, 11771–11782. The main novelty of the present study is the use of b-AsP as the other IR absorber which enables dependencies on both wavelength and linear polarization. This needs to be clarified, and the prior art properly acknowledged.

The authors have used the term 'twisted' to refer to the misalignment of the crystal structures between the bP and the bPAs layers. However, in 2D optoelectronics the term 'twisted' heterostructures has become synonymous with the formation of Moiré quantum materials. In the present study, the layers that are twisted with respect to each other are separated by a bulk MoS2 interlayer, and hence no such interaction occurs. As such, I suggest the Authors choose a different term to avoid confusing the readership— perhaps misaligned or right-angle alignment.

Minor comments:

The cartoon device structure in Fig 1d shows b-P and b-AsP layers as a bilayer and a trilayer respectively. However, presumably these layers are of bulk thickness i.e. more than 8 layers. I understand that this is just an illustrative diagram, but the authors should rework this figure (showing at least a few more layers) to avoid confusing the readership.

Related to the above point, the thickness of the b-P and b-AsP MWIR absorbers should be mentioned somewhere within the main text.

Whilst it is mentioned briefly within the text, for Figures 2 and 3 which deal with a 4-terminal device, I suggest the authors clarify which electrodes the measurements are taken between for each set of data presented. In the present version, it is not clear if each measurement is taken with the same set of electrodes.

Some of the figures are of low resolution. For example, the TEM images in the SI.

Reviewer comments (second round of review):

Reviewer #1

(Remarks to the Author)

The authors have carefully revised the manuscript according to the reviewers suggestion and they have given reasonable response to the comments. Therefore, I think the manuscript is now suitable for publication in Nat Comm.

Reviewer #2

(Remarks to the Author)

The authors have answered my most of concerns, it can be considered for publication at present stage.

Reviewer #3

(Remarks to the Author)

(Reviewer 3) The Authors have addressed some of the comments made during review. However, the main comments need to be better addressed, particularly comments 1-3. At present the manuscript is unsuitable for publication in a high impact journal such as Nature Communications.

Reviewer 3, comment 1: the purpose of this comment was for the Authors to provide specific information within the main text so the readers could understand, and ideally reproduce, the calculation of specific detectivity, more explicitly. As such:

- The illumination power density from which the best detectivity was measured should be listed i.e. in $W/\mu m^2$.
- The areas of the device (i.e. in μm^2) should be listed. The use of different areas for the different bias conditions is very important and should also be mentioned.
- The bias conditions for each measurement/figure should be clearly labelled (i.e. in V).

In addition, the conditions of the new noise measurement appear to be misaligned with the measured detectivity conditions? i.e. noise was measured at 0.2 V, detectivity measured at 0.3V? this would result in an underestimation of the noise.

Related to the above point, the authors now have measured noise and shown it is dominated by $1/f$ noise. However, the specific detectivity presented in the main text and conclusion appears to be still calculated assuming ideal noise behavior (i.e. equation 2)? The presented detectivities should be replaced with those calculated using the measured $1/f$ dominated noise (i.e. the real noise). I understand that many studies of 2D material detectors still use the idealized noise in their detectivity calculations. At the very least, a detectivity value calculated using the real noise should be included and made clear in the main text. If the Authors insist on keeping the idealized detectivity, then every time it is listed, it should be stated that this is a known overestimation.

Reviewer 3, comments 2 and 3: These comments questioned the validity of claiming that polarisation and spectral information could be differentiated simultaneously in any real-world situation. That is, only under very simple controlled situations would the detector be able to discriminate polarisation and spectral information simultaneously. For example:

- Repasting from the original comments "based on Figure 3a, how would this detector discriminate between $3 \mu m$ and $5 \mu m$ linearly polarized light aligned to the AC direction of the bottom b-AsP absorber (given that the top b-P will not strongly respond to either of these)"
- Based on the new figure S10, how would the detector discriminate, using the subtraction method, that the difference is due to polarisation aligned to the bottom absorber or illumination at longer wavelength?

The above two are just two obvious scenarios, but it seems that the detector would not be able to discriminate between polarisation and spectral shifts in any mixed polarisation/ different spectral distribution scenario. Based on their response to comment 2, the Authors agree that this is not possible with the reported detector, but they have not clarified this in the manuscript. I suggest the Authors:

- Add a section discussing/acknowledging this limitation, i.e. that using this detector to simultaneously obtain spectral and polarisation information from an unknown illumination source would not be possible except in very simple scenarios, which

do not correspond to real-world situations. Instead, at least two such detectors (at different orientations) would be required for this.

- Remove/clarify any statements that insinuate that polarisation and spectral information can be discriminated simultaneously. Such as “Our research proposes a practical approach for detecting multi-dimensional optical information” and other similar comments.

Minor comments:

Reviewer 3, comment 4 (also comment 1 of reviewer 2). While the Authors have now added citations to various previous (very similar) black phosphorus/arsenic-based bias switchable detectors, they are vague and non-specific. For example, “In addition, bias-switched spectral response detectors based on 2D materials have been reported” – which includes reference 15, a study in which a bP/MoS₂/bPAs detector is demonstrated (i.e. a very similar structure). The Authors could do a better job of clarifying the novelty of their specific detector in the context of previous work as this lack of clarity will confuse the readership. I suggest the authors add a dedicated paragraph in their introduction talking about all the previous black phosphorus-based bias switchable spectral/polarisation detectors (including their own work), and highlight the gap that the present study addresses.

Reviewer comments (third round of review):

Reviewer #3

(Remarks to the Author)

The authors have addressed most of my comments. However, there is one significant error that remains.

In this revision, the authors have remeasured their current noise and recalculated detectivity. This time they measured noise at ± 0.1 V, instead of ± 0.2 V. However, despite the small difference in measurement conditions they measure a current noise which is 4-5 orders of magnitude lower than the previous measurement they reported in the earlier version of this manuscript (in the low frequency region). The remeasured noise is even a couple of orders of magnitude lower than the shot noise limit for these conditions (based on Figure 3b). To my understanding this is fundamentally impossible, hence, it is very likely there is some error in this measurement. This is leading to the calculation of a very high MWIR detectivity (close to the BLIP limit), and much higher than was reported in the original manuscript (i.e. much higher than the values other reviewers saw).

It is important to get these details correct, the 2D IR detector community is already under scrutiny for not taking such measurements properly – see for example: A. Rogalski, Matters Arising, Nature Nanotechnology, volume 17, pages 217–219 (2022) and other papers by Rogalski commenting on this issue.

Reviewer comments (fourth round of review):

Reviewer #3

(Remarks to the Author)

The statement "Figure R1 indicates that the noise current density of the device at frequencies of 10 kHz to 100 kHz is not significantly attenuated compared to low frequencies." should be removed. The noise current is >3 orders of magnitude higher at low frequencies, which is quite significant considering the impact it would have on detectivity.

Response Letter to Reviewers

Reviewer #1 (Remarks to the Author):

In this manuscript, the authors have developed an innovative twisted unipolar barrier photodetector composed of b-AsP/MoS₂/BP heterostructure for acquiring multi-dimensional optical information. For spectral detection, they employ a two-terminal barrier structure with precise band engineering to enable bias-switchable dual-band detection. For polarization detection, they utilize anisotropic semiconductor BP and b-AsP as the absorbing layers, separated by a MoS₂ barrier layer. Furthermore, during the process of stacking heterogeneous junctions, they twist the crystal orientations of BP and AsP at a specific angle, rendering the two absorbing layers sensitive to incident light with different polarization angles.

It is impressive enough that such a structure allows for dual-band detection and polarization detection simultaneously on only one pixel. I suggest to accept this manuscript. However, there are some issues that need to be addressed before the manuscript can be published.

We thank the reviewer for carefully reading the manuscript and providing several precious comments. And we appreciate the recognition of our research by the reviewer. We tried our best to improve the manuscript and made some changes to the new manuscript. These changes will not influence the content and framework of the paper. We would like to thank you again for your positive comments on our work. We have addressed the reviewer's comments carefully with the listed responses below:

1. In Figure 2, the authors used an 830 nm laser source for photocurrent mapping. It would be helpful to mention that previous studies have investigated the use of multilayer MoS₂ as photodetectors for light sources with a wavelength of

approximately 900 nm. Can the MoS₂ layer also absorb light at 830 nm?

Answer: This is an excellent question. Theoretically, the photogenerated electron hole pairs generated by the barrier layer MoS₂ can be collected. However, in the pBp barrier structure employed in this study, the absorption layer b-AsP is significantly thicker than the MoS₂ barrier layer. Consequently, the device is primarily influenced by the absorption layer, which absorbs photons and generates electron-hole pairs, even under visible light. Figures 2e and 2i illustrate that under bias voltages of 0.5 V and -0.5 V, the main absorption regions of the device are bP and b-AsP, respectively. This observation suggests that the key regions for producing photocurrent under operational bias voltages are situated within the bP or b-AsP absorption layers. Moreover, regardless of the bias direction, the electrodes at both ends of the device should theoretically collect the electron-hole pairs generated by MoS₂ to create photocurrent. However, when the device is operated at bias voltages of 0.5 V, 0.1 V, 0 V, -0.1 V, and -0.5 V (as depicted in Figures 2e, S4a, S4e, S4i, and 2i), it becomes apparent that the area outside regions I, II, and III in the yellow dashed region of Figure 2a (pure MoS₂ region) does not contribute to photocurrent generation, indicating that the role played by MoS₂ in this process is negligible.

2. Please explain why carriers are not collected from outside the studied area.

Answer: Thank you for your insightful question. The barrier detector demonstrates the selective response of the absorption layer on either side of the barrier by altering the bias polarity. As depicted in Figure 2, at a bias of 0.5 V, the generated holes in the b-AsP and bP layers move towards the right due to the bias effect. The barrier layer impedes the holes generated in the b-AsP layer, leading to their eventual recombination with electrons, thereby contributing no current. Conversely, the electron-hole pairs in the bP layer can be collected by the electrode, consequently contributing to the photocurrent. This mechanism elucidates why only the bP region in Figure 2e demonstrates photocurrent. Conversely, at a bias of -0.5 V, the scenario is reversed. The

holes in the bP layer are impeded, resulting in no photocurrent contribution from the bP layer. Meanwhile, the generated electron-hole pairs in the b-AsP layer can be collected, leading to the generation of photocurrent in the b-AsP region, as illustrated in Figure 2i. Furthermore, as discussed in the previous question, the pure MoS₂ region in the device does not contribute to photocurrent generation. Therefore, only the heterojunction regions generate photo-induced charge carriers and contribute to photocurrent.

3. In figure 3a, the photoresponse of the MWIR2 channel is very similar to that of the MWIR1 channel above the CO₂ line. Please explain why the light on the MWIR2 channel is not absorbed by the MWIR1 channel first?

Answer: This question is quite insightful. With conventional dual-band detectors, radiation is incident on the shorter-wavelength detector, while longer-wavelength radiation passes through to the next detector. Each layer absorbs radiation until its cutoff and is therefore transparent to longer wavelengths, which is then collected in subsequent layers. When designing the device, we hope that the bP layer can completely absorb infrared light below 4.2 μm . However, we found that even if the bP layer is 300 nm thick, it cannot achieve 100% absorption of incident light below 4.2 μm . Consequently, a significant amount of light is transmitted to b-AsP absorbing layer without being absorbed by bP. Typically, the optical overlap of traditional dual-band photodetectors does not support spectral resolution. However, in this case, we are able to differentiate the spectral information of MWIR1 and MWIR2 by leveraging the similar spectral response trend of bP and b-AsP in the MWIR1 range. When the normalized device detects incident light of an unknown wavelength, if the value of the device's channel 2 signal minus the channel 1 signal is zero, the incident light's wavelength falls within the MWIR1 range. If the value is greater than zero, the incident light is in the MWIR2 range (Supplementary Fig. 9c).

4. When the device is illuminated and driven by reverse bias, the bottom b-AsP layer acts as an absorber, while the top bP layer acts as a polarizer for polarization detection.

What effect does this have on the bottom b-AsP absorber layer?

Answer: Thank you for your insightful question. Similar to the previous question, bP is unable to absorb light completely. Some of the incident light on the device will transmit through bP and reach the subsequent absorbing layer of b-AsP. Therefore, when it comes to polarization-resolved detection, the bP layer cannot fully block the polarized light. If the twist angle between bP and b-AsP changes, it only results in a corresponding change in the phase of the photocurrent, as shown in Figure 4d, without affecting the magnitude of the photocurrent. This phenomenon has been confirmed in Ref.17[Adv. Mater. 2022, 34, 2203766], as shown in Figure R1. Originating from the natural linear dichroism, photocurrents at both forward and reverse bias dependent on the incident linear state of polarization could be described as a trigonometric Function Equations $I_{ph} \propto \cos 2\theta$, as shown in Figure 4b and 4c. Because of the consistency between anisotropic optical properties and photoresponse performance under the change of state of polarization, the phase difference of the bias selective photoresponse could be well controlled by the twisted angle ψ of b-AsP and bP.

[REDACTED]

Fig. R1 Theoretically study of twisted angle dependence of photoresponse scheme at forward and reverse bias[Ref.17: Adv. Mater. 2022, 34, 2203766].

5. Figure S11 shows that the responsivity depends on the optical power, but this is not discussed in the text. Please explain why the responsivity is not constant.

Answer: Thank you for your comment. As the light intensity is increased, the responsivity demonstrates a sublinear dependence on it. This reduction in

photoresponsivity can be explained in terms of trap states present either in 2D materials[Ref. 2: Nature Nanotechnology, 2013, 8, 497-501]. This is exacerbated by the high surface-to-volume ratio of the 2D materials. Under high illumination intensities the density of available states is reduced, resulting in saturation of the photoresponse. The presence of trap states can dramatically influence the dynamics of the 2D materials photodetector.

Reviewer #2 (Remarks to the Author):

In this work, the authors reported a twisted unipolar barrier photodetector based on bP/MoS₂/b-AsP heterojunctions, the device has a capability of bias-switchable dual-band detection due to the unipolar barrier structure. By employing the anisotropic semiconductors such as black phosphorus (bP) and black arsenic phosphorus (b-AsP) as two absorbing layers, the authors also achieved the polarization-detection capabilities with polarization ratio (PR) value of 24.7 and 11.8 for the bias voltages of 0.4 V and -0.4 V, respectively, in the mid-wave infrared range.

Although the as-fabricated twisted unipolar barrier photodetector is capable of acquiring multi-dimensional optical information (including intensity, spectrum and polarization), similar device architecture and results about this subject have been previously reported. For example, the unipolar-barrier van der Waals heterostructures based on twisted b-AsP/WS₂/b-AsP were reported [Adv. Mater. 2022, 34, 2203766], which also showcase a bias-switchable polarization-sensitive photo-detection with PR value up to 55 and 42 at bias of -0.1 and 0.1 V, respectively, at 4.6 μm, even outperforming the device described in this work. In addition, the bias-selectable and dual-band infrared detectors were also recently reported [ACS Nano 2023, 17, 12, 11771; Nano Lett. 2022, 22, 8, 3425]. Therefore, this work may not bring new conceptual or technological advance over the previous related reports. Given this, the reviewer cannot recommend it for publication in Nature Commun. The following are some concerns that need to be addressed.

Thank you for acknowledging our work. The next generation of optical sensing systems is moving towards integrated and intelligent development, enabling a single detector to capture multidimensional optical field information, including intensity, spectrum, polarization, and phase (as shown in Figure R2). By utilizing specific algorithms, the complementary advantages of information contained in various image data can be organically combined, resulting in the fusion of new, higher-dimensional image data.

This overturns the limitations of traditional optical system photodetectors, which can only detect one dimension of optical field information, such as polarization [Ref.17: Adv. Mater. 2022, 34, 2203766] or spectrum [Ref.15: ACS Nano 2023, 17, 12, 11771; Nano Lett. 2022, 22, 8, 3425]. Recent efforts have been made to obtain multi-dimensional optical information using photodetectors based on artificial metasurface structures and two-dimensional (2D) materials [Ref.19, 20 and 21]. While these works allow for the acquisition of polarization and wavelength information on a single pixel, they are limited to discerning discrete polarized light at pre-defined wavelengths, and the coupling of polarization and wavelength information prevents independent acquisition, thus restricting the range of applications.

Fig. R2 Photodetectors for multidimensional optical information acquisition

Current dual-band detectors, such as mercury cadmium telluride (HgTeCd or MCT)[Ref. 22, 23], quantum well infrared photodetectors (QWIPs)[Ref. 24, 25], and antimony-based type-II superlattices (T2SLs)[Ref. 26, 27], are fabricated using a multi-layered materials structure (see Figure R3). Radiation is incident on the shorter band absorber layer, with the longer wave radiation passing through to the next absorber layer. Each layer absorbs radiation up to its cutoff and is transparent to longer wavelengths, which are then collected in subsequent layers. In recent years, there have been numerous reports on dual-band detectors prepared using novel materials, such as the

sequential mode (bias-selectable) dual-band detector fabricated with HgTe quantum dots[Ref. 28] and the simultaneous mode dual-band detector fabricated with Si/MoS₂/bP hybrid heterostructures[Ref. 29]. However, the mature materials and quantum-dot dual-band IR photodetectors lack the capability of direct polarization detection. Integrating spectral and polarization detection in a single detector is highly challenging.

[REDACTED]

Fig. R3 Cross-section views of unit cells for various back-illuminated dual-band HgCdTe detector approaches. (a) Bias-selectable n-p-n structure reported by Raytheon [Proc. SPIE, 1994, 2274, 117–25]; (b) Simultaneous n-p-n design reported by Raytheon [J. Electron. Mater., 1998, 27, 747–51]

Therefore, by combining a structure that can achieve dual-band detection (sequential mode or simultaneous mode) with anisotropic 2D materials, a multi-dimensional optical information detector that can realize spectral detection and polarization detection on a single pixel can be achieved. In this study, we integrated anisotropic 2D materials with a unipolar barrier structure to create a sequential mode dual-band detector. Importantly, both wavelength ranges of the device can achieve polarization detection without the need for additional polarization components. Additionally, the dual-band detection and polarization detection functions are independent of each other, broadening the potential applications. Therefore, our research provides a feasible

approach for the future development of multi-dimensional light information acquisition.

In summary, in view of the progress we have made in multi-dimensional light information acquisition and the modifications to the article, we hope to gain further support and recognition from the reviewer, and we have addressed the reviewer's comments carefully with the listed responses below:

1. The authors fabricated a twisted unipolar barrier photodetector, what is the exact twist angle between bP and b-AsP? Whether the twist angle has influence on the performance?

Answer: Thank you for your comment. The device represented in Figure 4a features a 110° angle between the crystal orientations of bP and b-AsP. As long as the angle between the crystal orientations is not 0° , it does not affect the performance of the device. Any change in the angle only results in a corresponding change in the phase of the photocurrent, as illustrated in Figure 4d, without impacting the magnitude of the photocurrent. This phenomenon has been confirmed in previous study [Ref.17: Adv. Mater. 2022, 34, 2203766], as depicted in Figure R4. Arising from the natural linear dichroism, photocurrents at both forward and reverse bias, dependent on the incident linear state of polarization, could be described as a trigonometric function, $I_{ph} \propto \cos 2\theta$, as shown in Figure 4b and 4c. Due to the consistency between anisotropic optical properties and photoresponse performance under the change of state of polarization, the phase difference of the bias selective photoresponse could be well controlled by the twisted angle ψ of b-AsP and bP.

[REDACTED]

Fig. R4 Theoretically study of twisted angle dependence of photoresponse scheme at forward and reverse

bias[Ref.17: Adv. Mater. 2022, 34, 2203766].

2. How does the author get the energy band as shown in Figure 1a? The theoretical calculation or experiments should be performed, not just cited the previous reports.

Answer: Thank you for your comment. The band structure data in Figure 1a were provided by Professor Weida Hu, one of the authors of this article. Professor Hu's research group has previously used UPS and KPFM to analyze the energy band information of b-AsP, bP, and MoS₂ materials. The 2D material we are using is the same as the material previously studied by Professor Hu's research group. For detailed experimental information, please refer to Ref. 45 and 49.

3. On page 4: "while those from b-AsP cannot be collected due to the barrier (Figure 1b)." The barrier can block the transport of photo-generated holes, but the electrons from b-AsP should be collected by the external circuit. Does author explain this in more detail?

Answer: Thank you for raising this issue. We are sorry that we did not describe this issue in the article. The primary characteristic of a barrier structure is the presence of a unipolar barrier, which impedes the majority of charge carriers while allowing the minority carriers to move freely. This feature effectively suppresses the dark current generated from the contact layer. In this study, we have devised a pBp unipolar barrier structure that obstructs holes, while allowing electrons to move freely. As depicted in

Figure 1b, when the device operates under forward bias, bP functions as the absorber layer, while b-AsP serves as the contact layer. The holes generated in b-AsP are impeded by the barrier, and although the electrons produced in b-AsP can be collected by the external circuit, the obstructed holes in b-AsP ultimately recombine. This indicates that the obstructed holes in b-AsP will recombine with an equal number of electrons, thus the electron-hole pairs generated in b-AsP do not contribute to the current. The corresponding changes can be found in the revised manuscript on page 5, line 1-6.

4. On page 4: “the thickness of MoS₂ is confirmed to be less than 20 nm through atomic force microscopy (AFM). The thickness of the absorption layer can be adjusted as needed.” How does author optimize the thickness of MoS₂ and absorption layer? No related thickness-dependent experiments were found in the manuscript.

Answer: Thank you for raising this issue. We are sorry that we did not describe this issue in the article. Because among the devices we fabricated, the thickness of MoS₂ is up to 20 nm, and devices with thinner thicknesses can also realize bias-switchable to select different response channels. The focus of our research here is on the device to achieve spectral detection and polarization detection, and we have not discussed the optimal thickness of MoS₂ and absorption layer. We apologize for the misunderstanding caused by the lack of clarity in the manuscript. Typically, barrier layers are used to block majority carriers and provide good transmission capabilities for photogenerated minority carriers. For the thickness selection here, we refer to reports in similar barrier structure in Ref.10 and 17. Although we are also studying barrier structure devices here, what is different from previous reports is that we are more interested in its application in acquiring multi-dimensional optical information.

5. In the photocurrent mapping (Fig. 2e, i) under 830 nm laser, if the MoS₂ components contribute to the formation of photocurrent? Why it is or not.

Answer: This is an excellent question. Theoretically, the photogenerated electron hole

pairs generated by the barrier layer MoS₂ can be collected. However, in the pBp barrier structure employed in this study, the absorption layer b-AsP is significantly thicker than the MoS₂ barrier layer. Consequently, the device is primarily influenced by the absorption layer, which absorbs photons and generates electron-hole pairs, even under visible light. Figures 2e and 2i illustrate that under bias voltages of 0.5 V and -0.5 V, the main absorption regions of the device are bP and b-AsP, respectively. This observation suggests that the key regions for producing photocurrent under operational bias voltages are situated within the bP or b-AsP absorption layers. Moreover, regardless of the bias direction, the electrodes at both ends of the device should theoretically collect the electron-hole pairs generated by MoS₂ to create photocurrent. However, when the device is operated at bias voltages of 0.5 V, 0.1 V, 0 V, -0.1 V, and -0.5 V (as depicted in Figures 2e, S4a, S4e, S4i, and 2i), it becomes apparent that the area outside regions I, II, and III in the yellow dashed region of Figure 2a (pure MoS₂ region) does not contribute to photocurrent generation, indicating that the role played by MoS₂ in this process is negligible.

6. What is the noise current of the device? The noise spectral density should be measured.

Answer: Thank you for your kind reminder. We have supplemented the noise spectrum measurements for the b-AsP/MoS₂/bP device, as illustrated in Figure R5. It is evident that the noise of the device is determined by $1/f$ noise, regardless of whether it is at room temperature or liquid nitrogen temperature. The $1/f$ noise originates from the fluctuations of local electronic states resulting from static and dynamic disorder, including trapping/de-trapping and generation/recombination processes [Ref. 53: Science Advances. 2017, 3, e1700589]. Additionally, we have included the relevant data in Figure S13 of the revised manuscript.

Fig. R5 Noise spectrum measured for the device at room temperature and 77 K.

7. The resolution and contrast of photocurrent mapping in Fig. 4f-i is poor, what is the reason.

Answer: Thank you very much for your comment. The mapping plot in Figure 4f-i was generated using mid-wave infrared laser irradiation (4.6 μm), while Figure 2 utilized an 830 nm wavelength laser. In mapping scanning imaging systems, a smaller laser spot diameter leads to higher resolution. In this study, the mid-wave infrared laser had a relatively large spot diameter (approximately 20 μm), resulting in poorer imaging quality compared to the near-infrared laser (with a diameter of approximately 5 μm).

Reviewer #3 (Remarks to the Author):

The authors have fabricated a b-P/MoS₂/b-AsP MWIR photodetector and shown that they can detect both spectral and linear polarisation information from the incoming illumination source. The study builds on an existing body of literature from the authors and other groups; however, some aspects of the study are novel, and I believe it should eventually be published. There are several significant issues that need to be addressed beforehand, as detailed below.

We thank the reviewer for carefully reading the manuscript and providing several precious comments. And we appreciate the recognition of our research by the reviewer. We tried our best to improve the manuscript and made some changes to the new manuscript. These changes will not influence the content and framework of the paper. We would like to thank you again for your positive comments on our work. We have addressed the reviewer's comments carefully with the listed responses below:

Given the impressive room temperature detectivity values reported for unpolarised light (especially for the b-AsP on the bottom of the device underneath the thick b-P layer), the readership will want to see more information on the detectivity/responsivity calculations. For example, why do the blackbody responsivity values mentioned in the main text (~10 to 4 A/W) differ from the blackbody responsivity values provided in the SI Figure S11 (max 0.8 to 0.03 A/W)? at what power density were the champion detectivity results measured? What cut-off wavelengths are used to estimate the incident power from the blackbody for the b-P and b-AsP? What region of the device was used to estimate the active area? In addition, it would be better to measure the noise current directly.

We are grateful to the reviewer for their thorough review of the manuscript and their valuable comments. We apologize for not providing clear explanations of the calculation details for responsivity and detectivity.

In the main text, the responsivity was measured using device 5 (as shown in Figure 4a) under positive and negative 0.5 V bias conditions, with the black body temperature set at 450 °C. The detectivity in Figure S11 (now Figure S14 in the revised manuscript) was measured using device 4 (as shown in Figure S8) under bias conditions of plus and minus 0.3 V, with the blackbody temperature set at 800 °C.

In 2D material photodetectors, the detectivity of the device generally increases as the power density decreases [Ref. 2]. This can be observed in Figure S14, where the highest detectivity is achieved at the lowest power level.

According to Wien's displacement law $\lambda T = b$ [Figure S11b], the central wavelength of blackbody radiation at 450°C is 4 μm , while the central wavelength of blackbody radiation at 800°C is 2.7 μm .

The selection of the photosensitive area of the device is based on the discussion results in Figure 2. For instance, in Figure 2a, when the device operates in forward bias, the device exhibits photoresponse in the bP area (as shown in Figure 2e), so the photosensitive area of device 2 at this time is $A_2 + A_3$, where A_2 and A_3 are the areas of regions II and III, respectively. When the device 2 operates in reverse bias, the device displays photoresponse in the b-AsP area (as shown in Figure 2i). At this point, the photosensitive area of device 2 is $A_1 + A_3$, where A_1 is the area of region I.

We have included the noise spectrum measurements for the b-AsP/MoS₂/bP device, as depicted in Figure R6. It is evident that the device's noise is determined by 1/f noise, irrespective of whether it is at room temperature or liquid nitrogen temperature. The 1/f noise originates from the fluctuations of local electronic states resulting from static and dynamic disorder, including trapping/de-trapping and generation/recombination processes [Ref. 53: Science Advances. 2017, 3, e1700589]. Additionally, we have included the relevant data in Figure S13 of the revised manuscript.

Fig. R6 Noise spectrum measured for the device at room temperature and 77 K.

The ability to detect both spectral and polarisation information in a single detector is presented as an advantage in the present study. However, this may not translate to a practical advantage. For example, based on Figure 3a, how would this detector discriminate between 3 μm and 5 μm linearly polarised light aligned to the AC direction of the bottom b-AsP absorber (given that the top b-P will not strongly respond to either of these)? According to the subtraction method in the main text, this would be interpreted as light above 4.2 μm . Hence, while the detector can independently detect polarisation and spectral information, I don't think it can perform these simultaneously without a second detector/polariser. This should be clarified within the text.

Answer: Thank you for your insightful comment. It directly addresses the core issue. In conventional dual-band detectors, shorter-wavelength radiation is absorbed by the first detector, while longer-wavelength radiation passes through to the subsequent detector. Each layer absorbs radiation until its cutoff and is transparent to longer wavelengths, which are then collected in subsequent layers. In designing the device, our aim was for the bP layer to fully absorb infrared light below 4.2 μm , allowing for independent polarization and spectral detection. However, we discovered that even with a 300 nm thick bP layer, complete absorption of incident light below 4.2 μm could not be achieved. As a result, we had to resort to a compromise method of subtracting two

signals to distinguish different spectra. Further clarification can be found in the revised manuscript on page 8, lines 7-8.

In addition to the above comment, the spectra provided in Figure 3a represent the simplest extreme situation (i.e linear polarisation aligned to the two crystal axis). To support the claim that the detector can use a subtraction method (or similar) to provide spectral/polarisation information, the bias-switchable spectral behaviour should also be measured using an unpolarised source (or even as a function of linear polarisation angle). I suspect that such method would hold up well to this more realistic case. I suggest the authors perform such measurements if they want to make these claims.

Answer: We express our sincere gratitude to the reviewers for their valuable suggestions in improving the article. It is important to note that we utilized a non-polarized source to measure the bias-switchable spectral behavior, as shown in Figure R7, with the specific test configuration detailed in Figure S9a. For the polarization test, a ZnSe polarizer was placed in front of the sample. The effectiveness of observing the bias switching spectral behavior with non-polarized source measurements is evident. However, since we found the results of the non-polarized source test to be somewhat repetitive with the results of the polarized source test along the AC direction, we omitted them from the article, potentially leading to reader misinterpretation. To address this, we have included the results obtained from the non-polarized source in the supplementary material. Furthermore, we have incorporated the relevant data in Figure S10 of the revised manuscript.

Fig. R7 FTIR response spectrum with non-polarized source of the device.

The novelty of the study needs to be clarified and the existing literature acknowledged. For example, the concept of detecting linear polarisation using layers of anisotropic 2D materials stacked at right angle to one another in a bias-switchable form is presented as novel, however, this concept has been demonstrated previously. See for example Nature Photonics volume 12, pages 601–607 (2018) which uses a very similar b-P/MoS₂/b-P structure to detect MWIR linear polarisation. This similar earlier study is not mentioned in the manuscript, which may mislead the readership about the specific novelty of the work. Similarly, more recent studies exploring bias-switchable spectral response of b-P/MoS₂/other IR absorber devices have not been acknowledged, see for example: ACS Appl. Mater. Interfaces 2022, 14, 28, 32665–32674 and ACS Nano 2023, 17, 12, 11771–11782. The main novelty of the present study is the use of b-AsP as the other IR absorber which enables dependencies on both wavelength and linear polarization. This needs to be clarified, and the prior art properly acknowledged.

Answer: Thank you very much for your comment. Current dual-band detectors, such as mercury cadmium telluride (HgTeCd or MCT)[Ref. 22, 23], quantum well infrared photodetectors (QWIPs)[Ref. 24, 25], and antimony-based type-II superlattices (T2SLs)[Ref. 26, 27], are fabricated using a multi-layered materials structure (see Figure R8). The lattice mismatch issue during heteroepitaxial growth of these materials affects device performance. In recent years, there have been numerous reports on dual-color detectors prepared using novel materials without lattice mismatch problem, such as the sequential mode (bias-selectable) dual-band detector fabricated with HgTe

quantum dots[Ref. 28] and the simultaneous mode dual-band detector fabricated with Si/MoS₂/bP hybrid heterostructures[Ref. 29]. However, the mature materials and quantum-dot dual-band IR photodetectors lack the capability of direct polarization detection. Integrating spectral and polarization detection in a single detector is highly challenging. 2D materials have the potential to achieve this purpose. Layered 2D materials, with their unique physical properties, hold great promise for optoelectronic applications[Ref. 12,13]. These materials are composed of weak out-of-plane van der Waals (vdWs) bonds and lack dangling bonds on their surfaces after exfoliation. Consequently, different 2D materials can be combined to form vdWs heterojunctions, regardless of lattice mismatch, resulting in high-quality mutant heterojunctions[Ref. 16, 17]. Notably, photodetectors based on anisotropic 2D materials can achieve polarization detection without additional surface structures or external polarization optical elements[Ref. 24, 25, 26 and 27]. In addition, bias-switched spectral response detectors based on 2D materials have been reported[Ref. 14, 15]. Therefore, by combining a structure that can achieve dual-band detection (sequential mode or simultaneous mode) with anisotropic 2D materials, a multi-dimensional optical information detector that can realize spectral detection and polarization detection on a single pixel can be achieved. The corresponding changes can be found in the revised manuscript on page 3.

[REDACTED]

Figure.R8 Cross-section views of unit cells for various back-illuminated dual-band HgCdTe detector approaches.

(a) Bias-selectable n-p-n structure reported by Raytheon [Proc. SPIE, 1994, 2274, 117–25]; (b) Simultaneous n-p-n

The authors have used the term ‘twisted’ to refer to the misalignment of the crystal structures between the bP and the bPAs layers. However, in 2D optoelectronics the term ‘twisted’ heterostructures has become synonymous with the formation of Moiré quantum materials. In the present study, the layers that are twisted with respect to each other are separated by a bulk MoS₂ interlayer, and hence no such interaction occurs. As such, I suggest the Authors choose a different term to avoid confusing the readership—perhaps misaligned or right-angle alignment.

Answer: Thank you for your kind reminder. The term ‘twisted’ has been replaced with ‘misalignment’ in the revised manuscript. These modifications have been implemented without impacting the content and structure of the article.

Minor comments:

The cartoon device structure in Fig 1d shows b-P and b-AsP layers as a bilayer and a trilayer respectively. However, presumably these layers are of bulk thickness i.e. more than 8 layers. I understand that this is just an illustrative diagram, but the authors should rework this figure (showing at least a few more layers) to avoid confusing the readership.

Answer: Thank you for your kind reminder. The material thickness in the device structure schematic has been increased, as depicted in Figure 1d in the revised manuscript.

Related to the above point, the thickness of the b-P and b-AsP MWIR absorbers should be mentioned somewhere within the main text.

Answer: Thank you for your kind reminder. The thickness information of the material

has been mentioned in the revised manuscript on page 8, line 5-6.

Whilst it is mentioned briefly within the text, for Figures 2 and 3 which deal with a 4-terminal device, I suggest the authors clarify which electrodes the measurements are taken between for each set of data presented. In the present version, it is not clear if each measurement is taken with the same set of electrodes.

Answer: Thank you for your kind reminder. The electrodes utilized in the photocurrent mapping in Figure 2 are A1 and B2, and the relevant descriptions can be found in the revised manuscript on page 6, lines 23-24. The data in Figure 3 comprises multiple test processes, including FTIR spectral response measurement, blackbody radiation response measurement, time-resolved response measurement, and colorimetric temperature measurement. It is important to note that not all of these tests can be conducted with a single device. Therefore, we apologize for the unclear description of the test electrode in Figure 3.

Some of the figures are of low resolution. For example, the TEM images in the SI.

Answer: Thank you for addressing the shortcomings in the manuscript. We have uploaded a high-definition version of each picture in the manuscript separately, and we have added dotted lines and false colors to the TEM pictures in the SI to assist in identification.

Response Letter to Reviewers

Reviewer #1 (Remarks to the Author):

The authors have carefully revised the manuscript according to the reviewers suggestion and they have given reasonable response to the comments. Therefore, I think the manuscript is now suitable for publication in Nat Comm.

Reviewer #2 (Remarks to the Author):

The authors have answered my most of concerns, it can be considered for publication at present stage.

Reviewer #3 (Remarks to the Author):

(Reviewer 3) The Authors have addressed some of the comments made during review. However, the main comments need to be better addressed, particularly comments 1-3. At present the manuscript is unsuitable for publication in a high impact journal such as Nature Communications.

We express our gratitude to the reviewer for acknowledging that we have addressed certain previous concerns. We sincerely apologize for any shortcomings in our prior response. We have thoroughly considered the questions and concerns raised by the reviewers and have prepared a comprehensive point-by-point response. Our objective is to effectively address all uncertainties and ultimately secure the reviewer's support and acknowledgement.

Reviewer 3, comment 1: the purpose of this comment was for the Authors to provide specific information within the main text so the readers could understand, and ideally reproduce, the calculation of specific detectivity, more explicitly. As such:

- The illumination power density from which the best detectivity was measured should be listed i.e. in W/um².
- The areas of the device (i.e. in um²) should be listed. The use of different areas for the different bias conditions is very important and should also be mentioned.
- The bias conditions for each measurement/figure should be clearly labelled (i.e. in V).

Answer: Thank you for providing valuable editing suggestions to optimize our manuscript. We apologize for not answering your question clearly before. The newly submitted manuscript has been revised according to your suggestions. The detectivity was measured using a 500 °C blackbody radiation source at room temperature (20 °C), with the detector being 10 cm away from the blackbody radiation source. The effective incident power of the blackbody radiation can be calculated using the equation[*Sci. Adv.* 2022, 8, eabn1811]: $P_{bb} = \eta \frac{\sigma(T_b^4 - T_0^4)A_b A_D}{2\sqrt{2}\pi L^2}$, where $\eta = 0.7$ is the transmittance of Dewar window, σ is the Stefan-Boltzmann constant, T_b is the blackbody temperature, T_0 is the operating temperature of the photodetector, A_b is the area of the blackbody radiation aperture and L is the distance from the blackbody to the detector. A_D is the area of the active region of the detector (as shown in Figures 2e and 2f, the regions of forward and reverse bias responses for the device are II + III regions and I + III regions, respectively. Here, the active region of the detector under forward bias and reverse bias for our sample are 740 μm^2 and 1300 μm^2 , respectively). When the blackbody temperature is 500 ° C, the blackbody radiation power density at a distance of 10 cm from the blackbody radiation source can be calculated as 7.9×10^{-11} W μm^{-2} . The discussion on the illumination power density has been mentioned in the revised manuscript on page 9, line 25-27. The device response area under different bias conditions is also listed on page 9, line 21-24. The bias conditions for each measurement/figure should be clearly

labelled, as shown in Figure 3c, 3d, 3e and S13.

In addition, the conditions of the new noise measurement appear to be misaligned with the measured detectivity conditions? i.e. noise was measured at 0.2 V, detectivity measured at 0.3V? this would result in an underestimation of the noise.

Answer: Thank you for your comments. We have taken your comments into consideration and made the following revisions. Firstly, we have re-characterized the noise spectrum of the device at room temperature for both $V_d = 0.1$ V and $V_d = -0.1$ V, as illustrated in the Figure R1 (Figure S12 in the manuscript). Additionally, we have recalculated the detectivity of the device at both $V_d = 0.1$ V and $V_d = -0.1$ V. The details of the detectivity calculation can be found in the revised manuscript on page 10, line 9.

Figure R1. Noise spectrum measured for the device at room temperature.

Related to the above point, the authors now have measured noise and shown it is dominated by 1/f noise. However, the specific detectivity presented in the main text and conclusion appears to be still calculated assuming ideal noise behavior (i.e. equation 2)? The presented detectivities should be replaced with those calculated using the measured

1/f dominated noise (i.e. the real noise). I understand that many studies of 2D material detectors still use the idealized noise in their detectivity calculations. At the very least, a detectivity value calculated using the real noise should be included and made clear in the main text. If the Authors insist on keeping the idealized detectivity, then every time it is listed, it should be stated that this is a known overestimation.

Answer: We express our gratitude to the reviewer for providing valuable feedback regarding the calculation of detectivity. The noise spectrum of the device from 1 to 10^5 Hz at room temperature as shown in Figure R1. When the frequency exceeds 10^3 Hz, the noise behavior is characterized as $1/f$ noise. The low-frequency noise current, denoted as I_n , was estimated to be 3.2×10^{-15} A Hz^{-1/2} ($V_d = 0.1$ V) and 7.1×10^{-15} A Hz^{-1/2} ($V_d = -0.1$ V). Based on the data extracted from Fig. 3b, we can calculate the blackbody responsivity of the MWIR1 channel ($V_d = 0.1$ V) and MWIR2 channel ($V_d = -0.1$ V) to be 857 mA/W and 519 mA/W, respectively. Therefore, the noise equivalent power (NEP) at $T_b = 500$ ° C can be calculated as 3.7×10^{-15} W Hz^{-1/2} ($V_d = 0.1$ V) and 1.4×10^{-14} W Hz^{-1/2} ($V_d = -0.1$ V). Consequently, the blackbody detectivity of our photodetector could be estimated by $D^* = \frac{\sqrt{A_D \Delta f}}{NEP}$. Where A_D is the effective area of the photodetector, Δf is the bandwidth, R is the responsivity. The specific detectivity D^* of the MWIR1 photodetector and the MWIR2 photodetector can be calculated as 7.4×10^{11} cmHz^{1/2}W⁻¹ and 2.6×10^{11} cmHz^{1/2}W⁻¹. The details of the detectivity calculation can be found in the revised manuscript on page 10, line 9-14.

Reviewer 3, comments 2 and 3: These comments questioned the validity of claiming that polarization and spectral information could be differentiated simultaneously in any real-world situation. That is, only under very simple controlled situations would the detector be able to discriminate polarization and spectral information simultaneously. For example:

- Repasting from the original comments “based on Figure 3a, how would this detector discriminate between 3 μ m and 5 μ m linearly polarized light aligned to the AC direction

of the bottom b-AsP absorber (given that the top b-P will not strongly respond to either of these)”

- Based on the new figure S10, how would the detector discriminate, using the subtraction method, that the difference is due to polarisation aligned to the bottom absorber or illumination at longer wavelength?

Answer: We express our gratitude to the reviewers for their valuable insights and professional opinions.

To discriminate between 3 μm and 5 μm linearly polarized light aligned to the AC direction of the bottom b-AsP absorption layer, the device can be measured under both forward bias and reverse bias. If the device only produces a signal under reverse bias but not under forward bias, it indicates that the incident light is absorbed by the b-AsP layer rather than the bP absorption layer, allowing us to conclude that the incident light is 5 μm . Conversely, if the device produces signals under both forward bias and reverse bias, it suggests that the incident light can be absorbed by both the bP layer and the b-AsP layer, indicating that the incident light is 3 μm .

Figure S10 displays the Fourier transform infrared spectrum response obtained from a non-polarized light source. Since the response is generated by a non-polarized light source, there is no need to consider the polarization alignment. Any differences observed can solely be attributed to the longer wavelength of the illumination.

The above two are just two obvious scenarios, but it seems that the detector would not be able to discriminate between polarisation and spectral shifts in any mixed polarisation/ different spectral distribution scenario. Based on their response to comment 2, the Authors agree that this is not possible with the reported detector, but they have not clarified this in the manuscript. I suggest the Authors:

- Add a section discussing/acknowledging this limitation, i.e. that using this detector to simultaneously obtain spectral and polarisation information from an unknown

illumination source would not be possible except in very simple scenarios, which do not correspond to real-world situations. Instead, at least two such detectors (at different orientations) would be required for this.

- Remove/clarify any statements that insinuate that polarisation and spectral information can be discriminated simultaneously. Such as “Our research proposes a practical approach for detecting multi-dimensional optical information” and other similar comments.

Answer: Thank you to the reviewers for their valuable feedback. This manuscript proposes a novel method for achieving spectral and polarization detection on a single pixel. Reconstructing the spectral and polarization information of incident light with unknown characteristics can be challenging. Currently, deep learning is used to reconstruct this unknown information through extensive data training, which is also a future direction of our research.

The manuscript showcases the capability of our device to differentiate between linearly polarized light at various angles through the manipulation of bias voltage. However, it is important to note that the method described above for distinguishing polarization states is only suitable for light with a wavelength within the overlapping absorption region of the bP and b-AsP spectra. Specifically, it applies to linearly polarized incident light with a wavelength below 4.2 μm . For the identification of polarization states of light above 4.2 μm , at least two detectors are necessary to achieve accurate identification. This statement has been added to the manuscript page 13, line 29.

Any statements that polarization and spectral information can be distinguished simultaneously have been modified to emphasize that both spectral and polarization detection can be achieved on a single photodetector. For example, “Our research proposes a practical approach for detecting multi-dimensional optical information” has been changed to “Our research proposes a practical approach for integrating multi-band detection and polarization detection into a single photodetector”.

Minor comments:

Reviewer 3, comment 4 (also comment 1 of reviewer 2). While the Authors have now added citations to various previous (very similar) black phosphorus/arsenic-based bias switchable detectors, they are vague and non-specific. For example, “In addition, bias-switched spectral response detectors based on 2D materials have been reported” – which includes reference 15, a study in which a bP/MoS₂/bPAs detector is demonstrated (i.e. a very similar structure). The Authors could do a better job of clarifying the novelty of their specific detector in the context of previous work as this lack of clarity will confuse the readership. I suggest the authors add a dedicated paragraph in their introduction talking about all the previous black phosphorus-based bias switchable spectral/polarisation detectors (including their own work), and highlight the gap that the present study addresses.

Answer: Thanks to the reviewers for their revision comments. There have been reports on biased switchable spectral/polarization detectors based on black phosphorus (bP) or black arsenic phosphorus (b-AsP). These detectors utilize bP/MoS₂/bP heterojunction [Ref. 10] and b-AsP/WS₂/b-AsP heterojunction [Ref. 21] to switch bias polarity, thereby achieving different polarization angle resolution detection. The pnp vdWs heterojunction based on bP can achieve different spectrum detection by switching the bias voltage [Ref. 14]. Furthermore, acquiring spatial polarization information can be achieved by utilizing twisted bP vdWs stacking [Ref. 20]. Therefore, by utilizing the structure of a bias-switchable selective response channel and anisotropic 2D materials with different absorption cutoff wavelengths, we can develop a detector that obtains spectrum and polarization multi-dimensional light information on a single pixel. This dedicated paragraph has been added to the revised manuscript on page 3, line 24.

The gap addressed by this study has been discussed in the manuscript on page 4, namely, "In this work, we propose a misalignment unipolar barrier photodetector (MUBP) for acquiring multi-dimensional optical information. For spectral detection, we employ a

two-terminal barrier structure with precise band engineering to enable bias-switchable dual-band detection. Compared to three-terminal devices, the two-terminal devices eliminate the need for an additional common electrode, resulting in simpler device structure and fabrication processes, higher fill factor, and improved device resolution. It is important to note that the device, based on barrier structures, represents a new type of photodetector beyond the traditional pn junction, which has been extensively investigated in recent years for high-performance or multi-band IR detection. The unipolar barrier effectively blocks majority carrier transport while allowing minority carrier transport, leading to suppressed dark current without impeding photocurrent. For polarization detection, we utilize anisotropic semiconductor bP and b-AsP as the absorbing layers, separated by a MoS₂ barrier layer. Furthermore, during the process of stacking heterogeneous junctions, we twist the crystal orientations of bP and b-AsP at a specific angle, rendering the two absorbing layers sensitive to incident light with different polarization angles.”

REVIEWER COMMENTS

Reviewer #3 (Remarks to the Author):

The authors have addressed most of my comments. However, there is one significant error that remains.

In this revision, the authors have remeasured their current noise and recalculated detectivity. This time they measured noise at ± 0.1 V, instead of ± 0.2 V. However, despite the small difference in measurement conditions they measure a current noise which is 4-5 orders of magnitude lower than the previous measurement they reported in the earlier version of this manuscript (in the low frequency region). The remeasured noise is even a couple of orders of magnitude lower than the shot noise limit for these conditions (based on Figure 3b). To my understanding this is fundamentally impossible, hence, it is very likely there is some error in this measurement. This is leading to the calculation of a very high MWIR detectivity (close to the BLIP limit), and much higher than was reported in the original manuscript (i.e. much higher than the values other reviewers saw).

It is important to get these details correct, the 2D IR detector community is already under scrutiny for not taking such measurements properly – see for example: A. Rogalski, Matters Arising, Nature Nanotechnology, volume 17, pages 217–219 (2022) and other papers by Rogalski commenting on this issue.

Answer: We appreciate your approval of most of the revisions we made previously. We are also particularly grateful for your careful suggestion on the noise measurements in this work. We apologize for the mistakes made in the noise testing. Therefore, after troubleshooting potential setup issues, we re-characterized the noise current density of the device, as depicted in Figure R1. The trend of the noise current density in this iteration is consistent with the previously revised manuscript. The noise current density was measured using a preamplifier (SR570) and dynamic signal analyzer (Keysight 35670A) in the vacuum Dewar at room temperature.

Figure R1. Noise spectrum measured for the device at room temperature

We once again carefully studied Professor Rogalski's article (thanks to the reviewer for the recommendation) and recalculated the device's detectivity based on the noise spectrum. Figure R1 indicates that the noise current density of the device at frequencies of 10 kHz to 100 kHz is not significantly attenuated compared to low frequencies. Since our device bandwidth is 700 kHz (as shown in Figure 3c), we calculated the detectivity using the noise and response rate of the device at 10 kHz to 100 kHz. The noise current, denoted as I_n , was estimated to be $1 \times 10^{-13} \text{ A Hz}^{-1/2}$ ($V_d = 0.1 \text{ V}$) and $3 \times 10^{-13} \text{ A Hz}^{-1/2}$ ($V_d = -0.1 \text{ V}$). The blackbody responsivity R_{bb} was determined by $R_{bb} = I_{ph}/P_{bb}$. Based on the data extracted from Figure 3b, we calculated the blackbody responsivity of the MWIR1 channel ($V_d = 0.1 \text{ V}$) and MWIR2 channel ($V_d = -0.1 \text{ V}$) to be 857 mA/W and 519 mA/W, respectively.

To calculate the specific detectivity D^* , we utilized the equation

$$D^* = \frac{\sqrt{A_D \Delta f}}{NEP} = \frac{R \sqrt{A \Delta f}}{I_n}$$

where A_D is the effective area of the photodetector, and R is the responsivity. Using this formula, we calculated the specific detectivity D^* of the MWIR1 and the MWIR2 photodetector to be $2.3 \times 10^{10} \text{ cmHz}^{1/2} \text{ W}^{-1}$ ($V_d = 0.1 \text{ V}$) and $3 \times 10^{10} \text{ cmHz}^{1/2} \text{ W}^{-1}$ ($V_d = -0.1 \text{ V}$).

We have updated the above revisions in the revised manuscript (page 10, lines 2-15). We hope that you will approve with our modifications.

REVIEWER COMMENTS

Reviewer #3 (Remarks to the Author):

The statement "Figure R1 indicates that the noise current density of the device at frequencies of 10 kHz to 100 kHz is not significantly attenuated compared to low frequencies." should be removed. The noise current is >3 orders of magnitude higher at low frequencies, which is quite significant considering the impact it would have on detectivity.

Answer: Thank you for your kind reminder. The statement "Figure R1 indicates that the noise current density of the device at frequencies of 10 kHz to 100 kHz is not significantly attenuated compared to low frequencies." has been removed. We have updated the above revisions in the manuscript (page 10, lines 2-3).